# Critical role for isoprenoids in apicoplast biogenesis by malaria parasites

Megan Okada[1], Krithika Rajaram[2], Russell P Swift[2], Amanda Mixon[1], John Alan Maschek[3], Sean T Prigge[2], Paul A Sigala[1]*

[1]Department of Biochemistry, University of Utah School of Medicine, Salt Lake City, United States; [2]Department of Molecular Microbiology and Immunology, Johns Hopkins School of Public Health, Baltimore, United States; [3]Metabolomics Core, University of Utah Health Sciences Center, Salt Lake City, United States

**Abstract** Isopentenyl pyrophosphate (IPP) is an essential metabolic output of the apicoplast organelle in *Plasmodium falciparum* malaria parasites and is required for prenylation-dependent vesicular trafficking and other cellular processes. We have elucidated a critical and previously uncharacterized role for IPP in apicoplast biogenesis. Inhibiting IPP synthesis blocks apicoplast elongation and inheritance by daughter merozoites, and apicoplast biogenesis is rescued by exogenous IPP and polyprenols. Knockout of the only known isoprenoid-dependent apicoplast pathway, tRNA prenylation by MiaA, has no effect on blood-stage parasites and thus cannot explain apicoplast reliance on IPP. However, we have localized an annotated polyprenyl synthase (PPS) to the apicoplast. PPS knockdown is lethal to parasites, rescued by IPP and long- ($C_{50}$) but not short-chain ($\leq C_{20}$) prenyl alcohols, and blocks apicoplast biogenesis, thus explaining apicoplast dependence on isoprenoid synthesis. We hypothesize that PPS synthesizes long-chain polyprenols critical for apicoplast membrane fluidity and biogenesis. This work critically expands the paradigm for isoprenoid utilization in malaria parasites and identifies a novel essential branch of apicoplast metabolism suitable for therapeutic targeting.

*For correspondence:
p.sigala@biochem.utah.edu

**Competing interest:** The authors declare that no competing interests exist.

## Editor's evaluation

This is an excellent, innovative and high quality study that reveals an essential role for isoprenoids within the *Plasmodium* apicoplast, and demonstrates a likely polyprenol synthase that is required for apicoplast biogenesis. This is an important finding for understanding apicoplast and isoprenoid biology in general, and is significant because synthesis of isoprenoids appears to be the only essential role for the apicoplast in asexual intraerythrocytic stages.

## Introduction

*Plasmodium falciparum* malaria parasites are single-celled eukaryotes that harbor a non-photosynthetic plastid organelle called the apicoplast which houses core metabolic pathways and is essential for parasite viability (*Ralph et al., 2004*). Because human cells lack this organelle and many of its constituent enzymes, the apicoplast has been viewed as a potentially rich source of new parasite-specific drug targets. However, cashing in on this potential has proved challenging since many apicoplast pathways, including heme (*Ke et al., 2014*; *Goldberg and Sigala, 2017*) and fatty acid synthesis (*Yu et al., 2008*; *Shears et al., 2015*), are dispensable during parasite infection of erythrocytes when all malaria symptoms arise. Multiple antibiotics, including doxycycline and clindamycin, block apicoplast biogenesis and inheritance and kill parasites, but their slow activity over several lifecycles has been a fundamental limitation to broad clinical application (*Dahl and Rosenthal, 2007*).

A key, essential function of the apicoplast is biosynthesis and export of the isomeric isoprenoid precursors, isopentenyl pyrophosphate (IPP), and dimethylallyl pyrophosphate (DMAPP), via the non-mevalonate/methylerythritol phosphate (MEP) pathway. IPP and DMAPP, which can be interconverted by an IPP isomerase, are critical for diverse cellular processes that include prenylation of proteins involved in vesicular trafficking, dolichol-mediated protein glycosylation, and biosynthesis of mitochondrial ubiquinone and heme A (*Guggisberg et al., 2014*; *Jomaa et al., 1999*; *van Dooren et al., 2006*; *Simão-Gurge et al., 2019*). Indeed, exogenous IPP is able to rescue parasites from lethal apicoplast dysfunction or disruption, highlighting the essential requirement for this isoprenoid precursor outside the apicoplast in blood-stage parasites (*Yeh and DeRisi, 2011*). Consistent with these critical cellular roles for IPP, the MEP pathway inhibitor fosmidomycin (FOS) kills parasites in the first lifecycle of treatment (*Jomaa et al., 1999*; *Yeh and DeRisi, 2011*; *Uddin et al., 2018*). This first-cycle FOS activity contrasts with the delayed, second-cycle death observed for *Plasmodium* parasites treated with antibiotics such as doxycycline and clindamycin that are thought to block translation of the 35 kb apicoplast genome and the predominantly organelle maintenance pathways it encodes. These contrasting kinetics have led to a prevailing view in the literature that essential apicoplast functions can be segregated into two general categories: (1) anabolic pathways that produce metabolites required outside the apicoplast and whose inhibition causes first-cycle parasite death and (2) housekeeping pathways that are only required for organelle maintenance and whose inhibition causes delayed, second-cycle defects (*Ramya et al., 2007*; *Kennedy et al., 2019b*; *Kennedy et al., 2019a*). Although this simple paradigm has been useful for conceptualizing general apicoplast functions, exceptions to this model have been reported (*Uddin et al., 2018*; *Amberg-Johnson et al., 2017*; *Boucher and Yeh, 2019*; *Okada et al., 2020*) and thus its general validity remains uncertain.

Since exogenous IPP rescues parasites from lethal apicoplast disruption (*Yeh and DeRisi, 2011*), isoprenoid biosynthesis has been thought to only serve essential roles outside this organelle (*Guggisberg et al., 2014*; *Uddin et al., 2018*; *Kennedy et al., 2019a*; *Amberg-Johnson et al., 2017*; *Gisselberg et al., 2018*; *Imlay and Odom, 2014*; *Gisselberg et al., 2013*). Indeed, *P. falciparum* expresses an essential polyprenyl synthase (PPS, PF3D7_1128400) that appears to localize to the cytoplasm and other cellular foci outside the apicoplast and mitochondrion (*Gabriel et al., 2015a*). The dual farnesyl/geranylgeranyl pyrophosphate synthase (FPPS/GGPPS) activity of this protein is critical for condensing isoprenoid precursors into longer polyprenyl-PP groups required for diverse cellular processes such as protein prenylation and dolichol synthesis (*Gisselberg et al., 2018*; *Gabriel et al., 2015a*; *No et al., 2012*). In addition, known prenyltransferases, which attach prenyl groups such as farnesyl pyrophosphate (FPP) and geranylgeranyl pyrophosphate (GGPP) to client proteins, are also cytoplasmic (*Imlay and Odom, 2014*).

In contrast to this prevailing paradigm, we have unraveled a novel essential arm of isoprenoid metabolism and utilization within the apicoplast and provide direct evidence that IPP and its condensation into downstream linear isoprenoids are required for apicoplast branching and inheritance by daughter merozoites. Genetic knockout of MiaA-dependent tRNA prenylation, the only previously predicted isoprenoid-dependent pathway in the apicoplast (*Ralph et al., 2004*), had no effect on blood-stage parasites, and thus MiaA cannot account for apicoplast dependence on IPP. However, we have localized a previously annotated PPS (PF3D7_0202700) (*Tonhosolo et al., 2005*) to the apicoplast and show that its conditional knockdown (KD) is lethal to parasites, can be rescued by IPP and long- but not short-chain polyprenols, and blocks apicoplast inheritance. We posit that this apicoplast PPS functions downstream of IPP synthesis to produce longer-chain isoprenoids essential for apicoplast membrane fluidity during organelle biogenesis. This discovery critically expands the paradigm for isoprenoid utilization in *P. falciparum*, identifies a potential new apicoplast drug target, and uncovers an organelle maintenance pathway whose inhibition causes first-cycle defects in apicoplast inheritance in contrast to delayed death-inducing antibiotics.

## Results

### Apicoplast elongation and branching require isoprenoid precursor synthesis

The *P. falciparum* literature has focused almost exclusively on the essential roles of isoprenoid metabolism outside the apicoplast (*Guggisberg et al., 2014*; *Yeh and DeRisi, 2011*; *Kennedy et al., 2019b*;

*Kennedy et al., 2019a*; *Gisselberg et al., 2018*; *Imlay and Odom, 2014*). Nevertheless, several prior studies reported that MEP pathway inhibitors such as FOS and MMV008138 blocked apicoplast elongation in lethally treated parasites, suggesting a possible role for IPP in apicoplast biogenesis (*Nair et al., 2011*; *Bowman et al., 2014*; *Goodman and McFadden, 2014*). These prior studies, however, could not rule out that defects in apicoplast development caused by MEP pathway inhibitors were due to non-specific effects from the pleiotropic cellular dysfunctions inherent to parasite death (*Gisselberg et al., 2018*). We revisited FOS inhibition of apicoplast biogenesis to further test and distinguish specific versus non-specific effects on organelle development.

We first tested the effect of 10 μM FOS (10× $EC_{50}$) on apicoplast elongation in synchronized cultures of two different parasite strains: D10 parasites expressing the apicoplast-targeted acyl carrier protein (ACP) leader sequence fused to GFP ($ACP_L$-GFP) (*Waller et al., 2000*) and a recently published NF54 parasite line (PfMev) that expresses $ACP_L$-GFP as well as heterologous enzymes that enable cytoplasmic synthesis of IPP from exogenous mevalonate precursor, independent of the apicoplast MEP pathway (*Swift et al., 2020b*). Consistent with prior reports (*Nair et al., 2011*; *Bowman et al., 2014*; *Goodman and McFadden, 2014*; *Howe et al., 2013*), we observed that synchronized ring-stage parasites treated with FOS developed into multinuclear schizonts but failed to elongate the apicoplast, which retained a focal, unbranched morphology in PfMev (*Figure 1A* and *Figure 1—figure supplement 1*) and D10 parasites (*Figure 1—figure supplements 2–3*). Although MEP pathway activity is detectable in ring-stage parasites (*Zhang et al., 2011*; *Cassera et al., 2004*), identical inhibition of apicoplast elongation in schizonts was observed if FOS was added to trophozoites 12 hr after synchronization (*Figure 1A* and *Figure 1—figure supplements 1–3*), suggesting continued reliance on de novo synthesis. In contrast to FOS treatment, parasites treated with lethal doses (10–100× $EC_{50}$) of drugs that target processes outside the apicoplast, including DSM1 (mitochondrial dihydroorotate dehydrogenase inhibitor) (*Phillips et al., 2008*), atovaquone (ATV, mitochondrial cytochrome *b* inhibitor) (*Fry and Pudney, 1992*), blasticidin-S (Blast-S, cytoplasmic translation inhibitor) (*Mamoun et al., 1999*), or WR99210 (WR, cytoplasmic dihydrofolate reductase inhibitor) (*Fidock and Wellems, 1997*), exhibited normal apicoplast biogenesis as they developed into schizonts, very similar to untreated parasites (*Figure 1B and C*, and *Figure 1—figure supplements 2–3*). These observations strongly suggest that defects in apicoplast elongation observed with FOS treatment are due to specific inhibition of MEP pathway activity rather than non-specific, secondary effects of parasite death.

Multiple studies have reported that FOS-treated parasites grow normally in the presence of exogenous IPP and do not show evidence of apicoplast loss (*Yeh and DeRisi, 2011*; *Uddin et al., 2018*; *Gisselberg et al., 2013*; *Swift et al., 2020b*), suggesting that IPP rescues apicoplast biogenesis from FOS-induced defects. To directly test this conclusion, we simultaneously treated synchronized rings with 10 μM FOS and either 200 μM IPP or 50 μM mevalonate (for PfMev parasites) and observed normal apicoplast elongation and branching in schizonts (*Figure 1A and B*, and *Figure 1—figure supplements 1–3*), consistent with a prior report (*Bowman et al., 2014*). These observations directly support the conclusion that apicoplast elongation requires isoprenoid synthesis.

To further test this conclusion via genetic disruption rather than pharmacological inhibition, we utilized a previously reported line of PfMev parasites in which the apicoplast-targeted deoxyxylulose-5-phosphate synthase (DXS), the first enzyme in the MEP isoprenoid synthesis pathway, had been genetically deleted (ΔDXS) (*Swift et al., 2020a*). These parasites require exogenous mevalonate to support cytoplasmic IPP synthesis, since they lack a functional apicoplast MEP pathway. In the presence of 50 μM mevalonate, ΔDXS parasites displayed normal apicoplast elongation and branching in schizonts. However, washing out mevalonate from ring-stage ΔDXS parasites to ablate IPP synthesis resulted in multinuclear schizonts with focal, unbranched apicoplast morphologies identical to those observed in the presence of FOS (*Figure 1D and E*, and *Figure 1—figure supplement 4*). These results strongly support the conclusion that apicoplast elongation and branching require IPP synthesis.

## Inhibition of isoprenoid synthesis prevents apicoplast inheritance by daughter parasites

To stringently test that IPP synthesis is required for apicoplast biogenesis, we next asked if FOS treatment prevented daughter parasites from inheriting the apicoplast, as predicted to occur if the apicoplast fails to elongate and divide in schizonts and as commonly observed for antibiotic inhibitors of apicoplast housekeeping pathways (*Uddin et al., 2018*; *Dahl et al., 2006*). Simultaneous treatment of

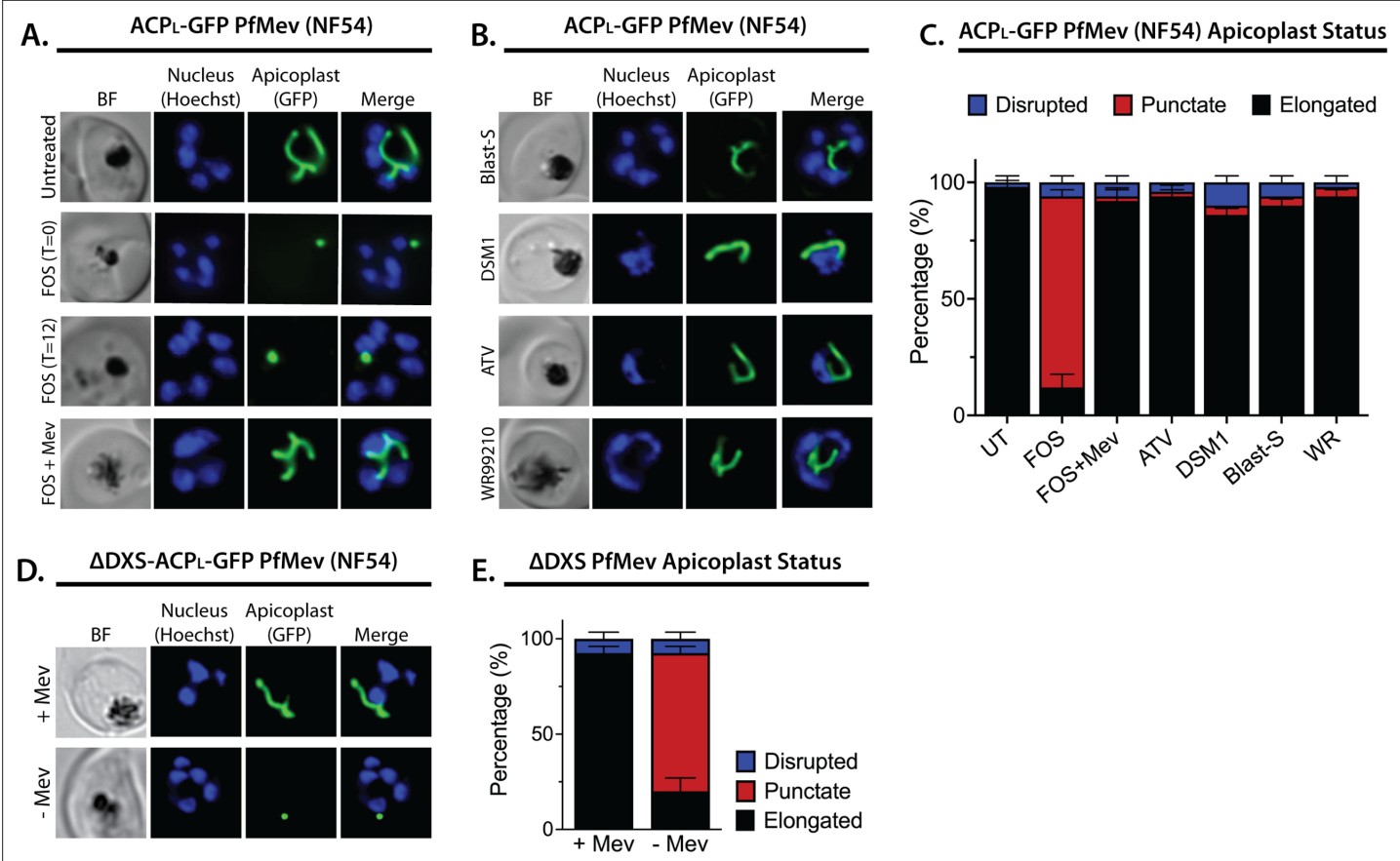

**Figure 1.** Inhibition of isoprenoid precursor biosynthesis specifically blocks apicoplast elongation and branching. Bright-field (BF) and fluorescent microscopy images of live NF54 PfMev parasites that were (**A**) untreated or treated with 10 µM fosmidomycin (FOS) in the absence or presence of 50 µM DL-mevalonate (Mev), or (**B**) treated with 6 µM blasticidin-S (Blast-S), 2 µM DSM1, 100 nM atovaquone (ATV), or 5 nM WR99210. (**C**) Population analysis of apicoplast morphology for 50 total parasites imaged for each condition in panels A and B from two independent experiments. Apicoplast morphologies were scored as punctate (focal), elongated, or disrupted (dispersed); counted; and plotted by histogram as the fractional population with the indicated morphology (UT = untreated). Error bars represent standard deviations from replicate experiments. (**D**) Live-cell imaging of ΔDXS PfMev parasites in the presence or absence of 50 µM Mev. (**E**) Population analysis of parasites imaged in panel D and performed as in panel C. In all experiments, synchronized ring-stage parasites were incubated with the indicated treatments for 36 hr prior to live-cell imaging. Parasite nuclei were visualized using 1 µg/mL Hoechst 33342. The parasite apicoplast was visualized using the ACP_L-GFP encoded by the PfMev line. Absolute parasite counts for microscopy experiments are shown in *Figure 1—source data 1*.

The online version of this article includes the following source data and figure supplement(s) for figure 1:

**Source data 1.** Absolute parasite counts for microscopy experiments.

**Figure supplement 1.** Additional epifluorescence images of PfMev ACP_L-GFP NF54 parasites treated with fosmidomycin (FOS) and other drugs at the indicated concentrations.

**Figure supplement 2.** Epifluorescence images and analysis of D10 ACP_L-GFP parasites treated with fosmidomycin (FOS) and other drugs.

**Figure supplement 3.** Additional epifluorescence images of D10 parasites treated with fosmidomycin (FOS) and other drugs.

**Figure supplement 4.** Additional epifluorescence images of ΔDXS PfMev parasites that were synchronized to ring stage with 5% D-sorbitol and incubated for 36 hr ± Mev hours prior to live-cell imaging.

ring-stage parasites with both FOS and IPP rescued growth defects and resulted in normal apicoplast elongation and division (*Figure 1A and B*, and *Figure 1—figure supplements 1–3*), as expected since IPP is the direct anabolic product of the MEP pathway specifically inhibited by FOS. Thus, concomitant treatment with IPP and FOS cannot distinguish whether MEP pathway inhibition prevents apicoplast inheritance by daughter parasites. To bypass this fundamental limitation, we devised the following alternative strategy.

The apicoplast begins to elongate near the onset of schizogony before branching and then dividing in late, segmenting schizonts (*Waller et al., 2000*; *van Dooren et al., 2005*). Despite manifesting defects in apicoplast elongation in early schizogony, FOS-treated parasites continue to divide nuclear DNA and transition into mature schizonts before stalling prior to segmentation into merozoites (*Figure 1A* and *Figure 1—figure supplements 1–3*; *Howe et al., 2013*). This observation suggested that defects in apicoplast biogenesis were not the immediate cause of parasite death in the current cell cycle and that such defects preceded a broader essential requirement for IPP outside the apicoplast in mature schizonts. Recent works suggest this broader essentiality to be IPP-dependent protein prenylation (*Kennedy et al., 2019a*; *Howe et al., 2013*). We therefore reasoned that if IPP supplementation were delayed until mid-schizogony, after the onset of apicoplast-elongation defects but before broader cellular death, it might be possible to rescue parasite viability without rescuing apicoplast biogenesis and thereby produce viable parasite progeny that lacked the intact apicoplast.

Synchronized ring-stage PfMev parasites were treated with 10 µM FOS for 48 hr, with 50 µM mevalonate added at 0, 30, 34, or 38 hr after synchronization. (*Figure 2A*). Parasites were allowed to expand for three subsequent cycles in 50 µM mevalonate, with growth monitored by flow cytometry. We observed a hierarchy of growth rescue by mevalonate, with full rescue (relative to no FOS treatment) of parasites supplemented with mevalonate at 0 hr post-synchronization and decreasing rescue for increasingly delayed supplementation at 30, 34, or 38 hr (*Figure 2B*), presumably due to fewer viable parasites surviving the initial cycle.

To assess and quantify apicoplast status in rescued parasites, we cloned out individual parasites at 60 hr post-synchronization in the second growth cycle. Apicoplast status in the resulting clones was determined by live parasite microscopy of organelle morphology, apicoplast genome PCR, and growth ± mevalonate. Although FOS-treated parasites supplemented simultaneously with mevalonate showed no evidence for apicoplast loss in clonal progeny, a fraction of clonal parasites derived from delayed mevalonate rescue showed clear signs of apicoplast loss, including a dispersed apicoplast ACP$_L$-GFP signal, loss of the apicoplast genome, and growth dependence on exogenous mevalonate (*Figure 2C* and *Figure 2—figure supplements 1–4*). The fraction of clonal parasites with a disrupted apicoplast increased from 10% in parasites supplemented with mevalonate at 30 hr to over 80% in parasites supplemented at 38 hr (*Figure 2D*). These results provide direct evidence that inhibiting IPP synthesis alone is sufficient to block apicoplast biogenesis and prevent organelle inheritance by daughter parasites.

## The MiaA pathway for apicoplast tRNA prenylation is dispensable for blood-stage parasites

Why do apicoplast elongation and branching require IPP synthesis? Currently, the only predicted isoprenoid-dependent metabolic pathway in the apicoplast is tRNA prenylation by MiaA (*Ralph et al., 2004*; *Imlay and Odom, 2014*), which catalyzes the attachment of a dimethylallyl group to the N$^6$ moiety of adenine at position 37 of certain tRNAs (*Persson et al., 1994*). DMAPP is produced in tandem with IPP in the terminal enzymatic step of the MEP pathway and can be interconverted with IPP by an IPP/DMAPP isomerase (*Guggisberg et al., 2014*; *Yeh and DeRisi, 2011*). Prenylation of A37 is often accompanied by methylthiolation by the radical SAM enzyme, MiaB (*Esberg et al., 1999*). Although genes encoding MiaA (PF3D7_1207600) and MiaB (PF3D7_0622200) are annotated in the *P. falciparum* genome and MiaA protein has been detected by mass spectrometry (MS) in the apicoplast-specific proteome (*Boucher et al., 2018*), neither protein has been studied biochemically in parasites. Nevertheless, both proteins are predicted to be non-essential for blood-stage *Plasmodium* based on genome-wide knockout (KO) studies in *Plasmodium berghei* (*Bushell et al., 2017*) and *P. falciparum* (*Zhang et al., 2018*).

To directly test whether MiaA function is essential for *P. falciparum* parasites and can account for apicoplast dependence on isoprenoid synthesis, we used CRISPR/Cas9 to target MiaA for gene disruption by double-crossover homologous recombination (*Figure 3—figure supplement 1*). PfMev parasites were transfected and selected in the presence of 50 µM mevalonate to ensure that parasites would remain viable even if deletion of MiaA resulted in apicoplast disruption. Parasites that had integrated the KO plasmid returned from transfection, and loss of the MiaA gene was confirmed by genomic PCR (*Figure 3—figure supplement 1*). The ΔMiaA parasites grew equally well in the presence or absence of mevalonate and grew indistinguishably from the parental PfMev parasites

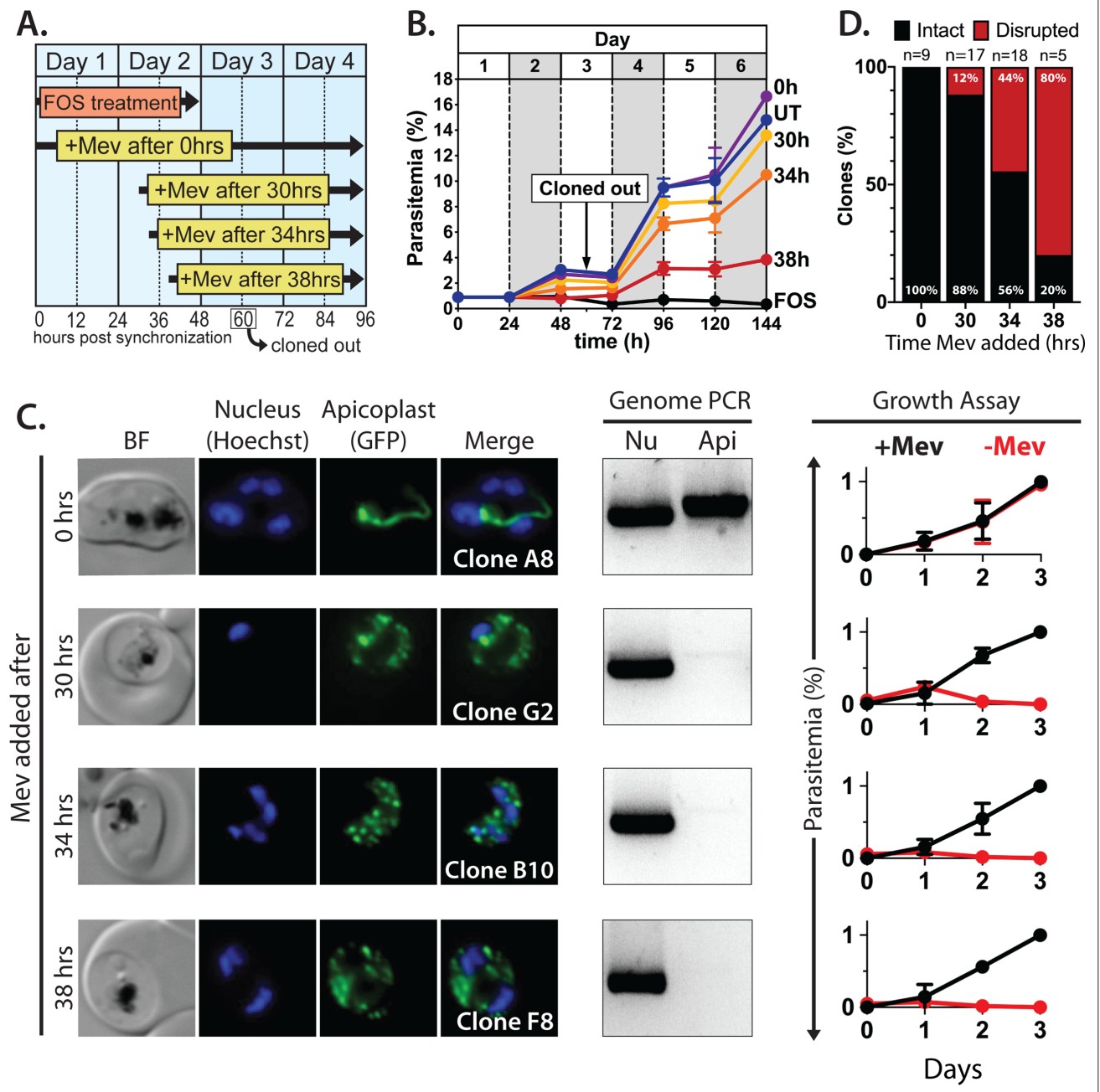

**Figure 2.** Inhibiting isoprenoid precursor biosynthesis prevents apicoplast inheritance by daughter parasites. (**A**) Schematic summary of delayed mevalonate rescue of fosmidomycin (FOS) treatment. PfMev parasites were synchronized with 5% D-sorbitol and cultured in 10 µM FOS (washed out after 48 hr in second-cycle rings) without or with addition of 50 µM DL-mevalonate (Mev) at 0, 30, 34, or 38 hr after synchronization. Clonal parasites from all growth conditions were isolated at 60 hr post-synchronization by limiting dilution and growth in 50 µM Mev. (**B**) Parasite growth was monitored for 6 days by flow cytometry using acridine orange staining (FOS = treated only with FOS, UT = untreated, 0–38 hr time delay of Mev addition after synchronization and initiation of FOS treatment). (**C**) Bright-field (BF) and fluorescence microscopy images of representative live clonal parasites with disrupted apicoplast (if observed) isolated after 60 hr of growth under the conditions described in panel A. Images of all clones are shown in **Figure 2— figure supplements 1–4**. Parasite nuclei were visualized using 1 µg/mL Hoechst 33342. To the right of each clonal image panel is a gel image showing the result of PCR analysis to amplify a (Nu) nuclear (PPS, PF3D7_0202700) and (Api) apicoplast (SufB, PF3D7_API04700) gene and a growth assay to monitor the ability of each indicated clone to grow in the presence or absence of 50 µM Mev. Data points are the average± SD of three biological replicates and were normalized to the parasitemia on day 3 of growth in +Mev conditions. (**D**) Graphical representation of the number of clones isolated under each growth condition and the clonal percentage with an intact or disrupted apicoplast (determined by microscope analysis of ACP_L-GFP signal and genomic PCR).

*Figure 2 continued on next page*

*Figure 2 continued*

The online version of this article includes the following source data and figure supplement(s) for figure 2:

**Source data 1.** Uncropped gel images of clonal PCR analyses.

**Figure supplement 1.** Epifluorescence microscopy images of clonal parasites isolated after fosmidomycin (FOS) treatment and rescue by mevalonate addition at 0 hr after synchronization.

**Figure supplement 2.** Epifluorescence microscopy images of clonal parasites isolated after fosmidomycin (FOS) treatment and rescue by mevalonate addition at 30 hr after synchronization.

**Figure supplement 3.** Epifluorescence microscopy images of clonal parasites isolated after fosmidomycin (FOS) treatment and rescue by mevalonate addition at 34 hr after synchronization.

**Figure supplement 4.** Epifluorescence microscopy images of clonal parasites isolated after fosmidomycin (FOS) treatment and rescue by mevalonate addition at 38 hr after synchronization.

(*Figure 3A*). The presence of an intact apicoplast was confirmed by genomic PCR analysis and live parasite microscopy (*Figure 3B* and *Figure 3—figure supplement 1*). These results indicate that MiaA is dispensable for blood-stage parasites and that deletion of this gene does not affect apicoplast biogenesis. Therefore, loss of function of MiaA, the only predicted isoprenoid-dependent pathway in the apicoplast, cannot account for apicoplast dependence on IPP synthesis, suggesting an alternative role for IPP in organelle elongation.

## Apicoplast biogenesis requires polyprenyl isoprenoid synthesis

Except for MiaA-catalyzed tRNA prenylation, all proposed roles for isoprenoids in *Plasmodium* parasites require head-to-tail condensation of DMAPP (5 carbons) and one or more IPP subunits (5 carbons) to form longer-chain isoprenoids, starting with formation of geranyl pyrophosphate (GPP, 10 carbons), FPP (15 carbons), and GGPP (20 carbons) (*Guggisberg et al., 2014*; *Imlay and Odom, 2014*). Recent studies reported that 5 µM geranylgeraniol (GGOH, the alcohol precursor of GGPP)

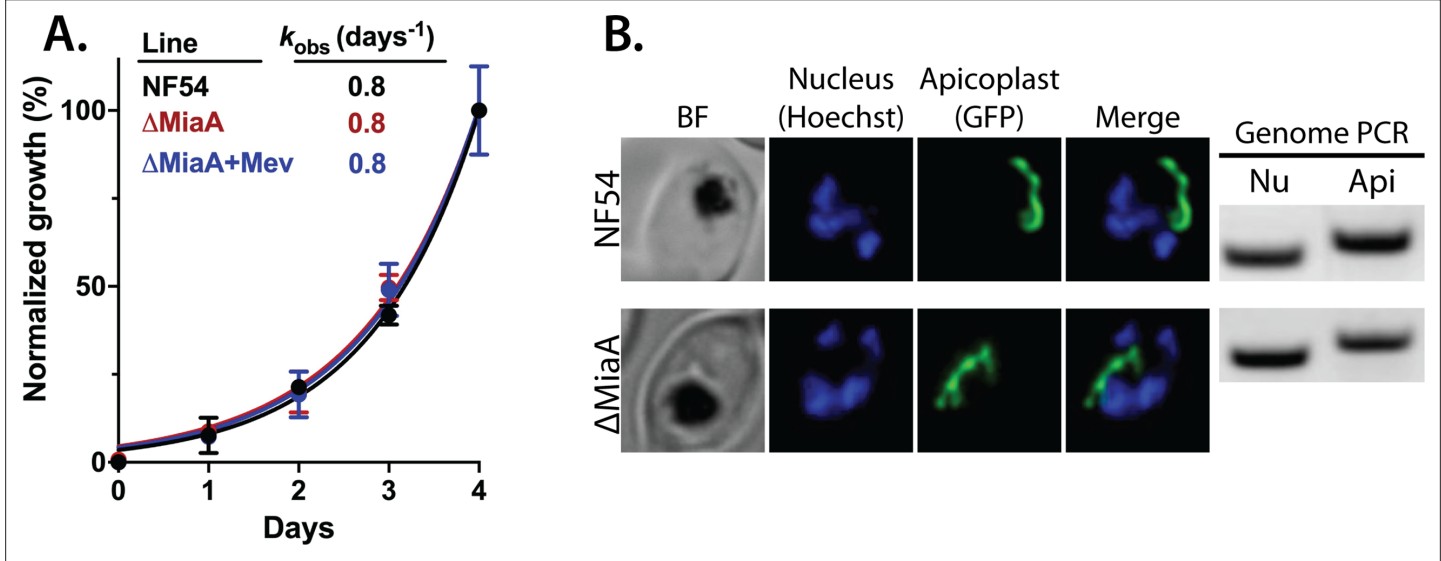

**Figure 3.** Genetic disruption of MiaA has no effect on parasite growth or apicoplast biogenesis. (**A**) Growth analysis indicates that parental PfMev NF54 parasites and ΔMiaA parasites cultured in the absence or presence of 50 µM Mev grow indistinguishably with identical rate constants ($k_{obs}$) for asynchronous culture expansion. Parasitemia values for each sample are the average ± SD of three biological replicates and were normalized to the parasitemia on day 4 and fit with an exponential growth model. (**B**) Live parasite imaging and genomic PCR analysis indicate normal apicoplast morphology and retention of the apicoplast genome in parental PfMev and ΔMiaA parasites. BF = bright field, Nu = nuclear gene (LDH, PF3D7_1324900), and Api = apicoplast gene (SufB, PF3D7_API04700).

The online version of this article includes the following source data and figure supplement(s) for figure 3:

**Source data 1.** Uncropped gel images of PCR analyses of nuclear and apicoplast genomes.

**Source data 2.** Uncropped gel images of PCR analysis to confirm disruption of MiaA gene.

**Figure supplement 1.** PCR genotyping of PfMev ΔMiaA parasites and additional epifluorescence images of apicoplast morphology.

can provide short-term (~1 cycle) rescue of parasite death due to treatment with FOS or indolmycin, an apicoplast tryptophan tRNA synthetase inhibitor (*Kennedy et al., 2019a*; *Howe et al., 2013*). Based on these reports, we hypothesized that the dependence of apicoplast biogenesis on IPP might reflect a requirement for longer-chain isoprenoids such that farnesol (FOH), GGOH, and/or longer-chain polyprenols might rescue the apicoplast branching defects caused by 10 µM FOS.

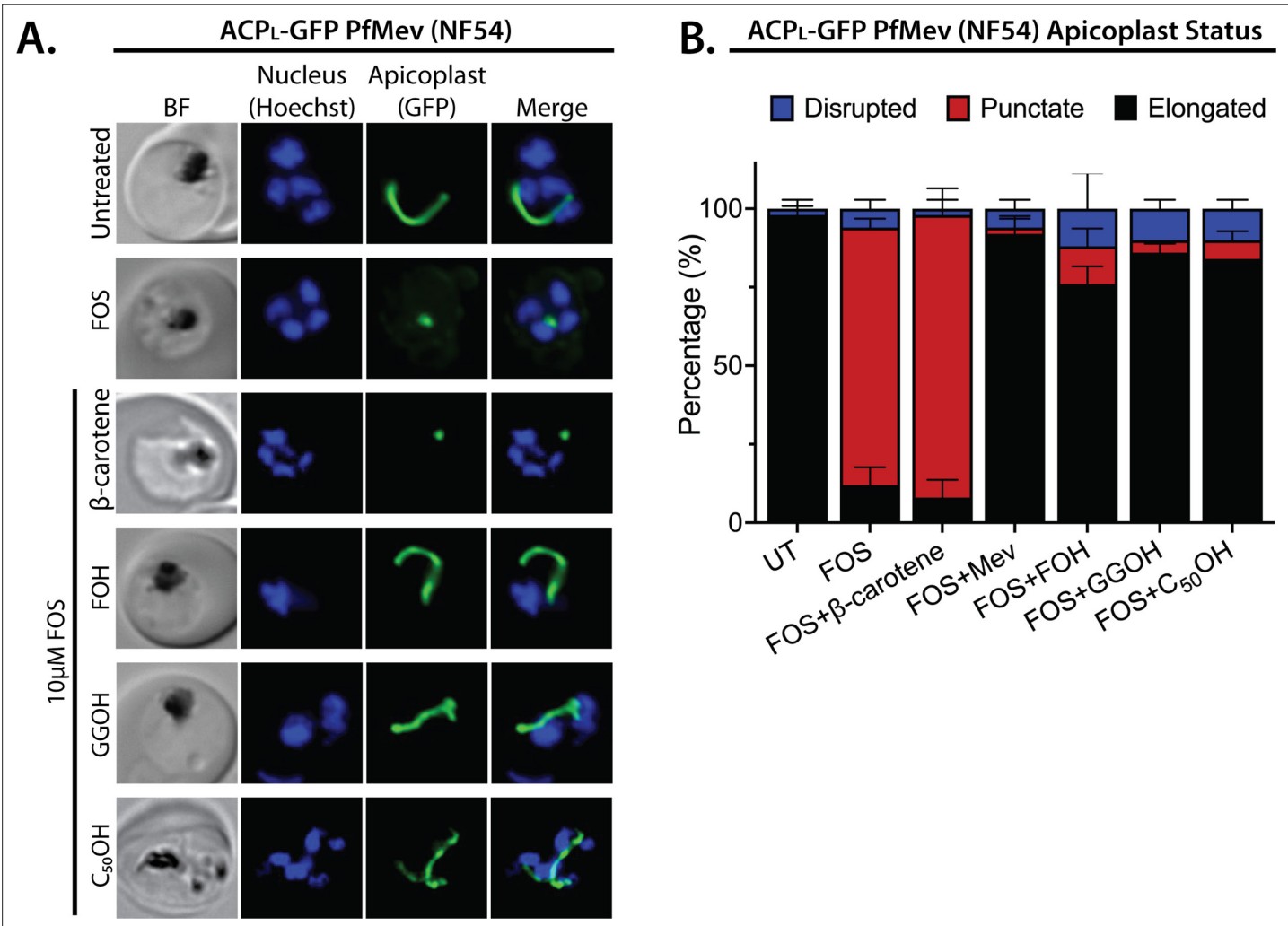

**Figure 4.** Apicoplast biogenesis requires linear polyprenyl isoprenoid synthesis. (**A**) Five µM farnesol (FOH), geranylgeraniol (GGOH), or decaprenol (C$_{50}$-OH), but not β-carotene rescues apicoplast biogenesis from inhibition by 10 µM fosmidomycin (FOS) in PfMev parasites. Synchronized ring-stage parasites were incubated with the indicated treatments for 36 hr and imaged by bright-field (BF) or fluorescence microscopy, with visualization of parasite nuclei by Hoechst staining and the apicoplast by ACP$_L$-GFP signal. (**B**) Population analysis of apicoplast morphology for 50 total parasites imaged for each condition in panel A from two independent experiments. Apicoplast morphologies were scored as punctate (focal), elongated, or disrupted (dispersed); counted; and plotted by histogram as the fractional population with the indicated morphology (UT = untreated). Error bars represent standard deviations from replicate experiments.

The online version of this article includes the following source data and figure supplement(s) for figure 4:

**Source data 1.** Absolute parasite counts for microscopy experiments.

**Figure supplement 1.** 5 µM geranylgeraniol (GGOH) but not farnesol (FOH) partially rescues parasite growth from inhibition by 10 µM fosmidomycin (FOS) in continuous-growth assays with PfMev parasites.

**Figure supplement 2.** Additional epifluorescence microscopy images of PfMev parasites treated with fosmidomycin (FOS) and farnesol (FOH), geranylgeraniol (GGOH), C$_{50}$-OH, or β-carotene.

**Figure supplement 3.** Epifluorescence microscopy images of D10 ACP$_L$-GFP parasites treated with fosmidomycin (FOS) and farnesol (FOH), geranylgeraniol (GGOH), C$_{50}$-OH, or β-carotene.

We treated synchronized NF54 and D10 parasites with both 10 µM FOS and 5 µM of either FOH or GGOH. Consistent with prior reports, 5 µM GGOH but not FOH partially rescued parasite growth from inhibition by FOS and enabled culture expansion into a second growth cycle (*Figure 4—figure supplement 1*). Nevertheless, both GGOH and FOH rescued apicoplast elongation and branching defects in schizonts when added simultaneously with FOS to synchronized rings (*Figure 4* and *Figure 4—figure supplements 2 and 3*). We extended these rescue experiments to include 5 µM decaprenol (50 carbons) and also observed rescue of apicoplast branching from FOS-induced defects. However, 5 µM β-carotene, which is a nonlinear carotenoid hydrocarbon derived from eight prenyl groups (40 carbons), did not rescue apicoplast biogenesis from inhibition by FOS (*Figure 4* and *Figure 4—figure supplements 2 and 3*). Although it is possible that β-carotene is not taken up efficiently into the apicoplast, rescue by decaprenol, which is similar in size and hydrophobicity to β-carotene, suggests that apicoplast biogenesis specifically requires synthesis of linear polyprenols containing three or more prenyl groups. This hypothesis is further supported by additional results described in the next two sections.

## Localization of an annotated PPS to the apicoplast

Iterative condensation of DMAPP with IPP subunits to form FPP, GGPP, and longer polyprenyl-PPs requires the function of a polyprenyl synthase (PPS). This family of enzymes uses a conserved dyad of DDXXD residues positioned near the protein surface of the active site binding pocket to coordinate $Mg^{2+}$ ions that bind the pyrophosphate headgroup of DMAPP, GPP, or FPP and position its allylic head relative to the vinyl tail of the IPP subunit (*Kellogg and Poulter, 1997*). Condensation of the two substrates via electrophilic alkylation elongates the nascent isoprenoid chain into the protein interior. Two amino acids just upstream of the first DDXXD motif determine the length of the resulting prenyl chain by forming a hydrophobic 'floor' that gates the depth of the protein interior. Indeed, dedicated FPPS enzymes feature an amino acid floor comprised of sequential Phe-Phe residues just upstream of the first DDXXD motif that sterically block synthesis of products longer than FPP (*Poulter, 2006*; *Thulasiram and Poulter, 2006*). Sequence variations that replace just the more N-terminal Phe or both Phe-Phe groups with smaller residues (e.g., Ala or Ser) open up and extend the binding pocket and enable synthesis of GGPP or longer polyprenyl-PPs up to 14 isoprene units, respectively (*Tarshis et al., 1996*).

A BLAST search of the *P. falciparum* genome with the sequence of the well-studied chicken FPP synthase (Uniprot P08836) revealed two parasite orthologs (PF3D7_1128400 and PF3D7_0202700) that retain the DDXXD dyads and other conserved sequence features expected of a PPS (*Figure 5A* and *Figure 5—figure supplement 1*). The best studied of these synthases is the dual-functional enzyme, PF3D7_1128400, which shares 34% sequence identity with avian FPPS and has been reported to catalyze formation of both FPP and GGPP (*Gabriel et al., 2015a*; *Jordão et al., 2013*; *Artz et al., 2011*). Consistent with its ability to synthesize GGPP as the terminal product, PF3D7_1128400 has sequential Ser-Phe residues just upstream of the first DDXXD motif (*Figure 5A*; *Jordão et al., 2013*; *Artz et al., 2011*). This enzyme is reported to be essential based on inhibitor (*Gisselberg et al., 2018*; *No et al., 2012*) and gene-disruption studies in *P. berghei* (*Bushell et al., 2017*) and *P. falciparum* (*Zhang et al., 2018*) and is thought to synthesize the FPP and GGPP required for broad parasite isoprenoid metabolism, including protein prenylation and synthesis of dolichols, ubiquinone, and heme A (*Guggisberg et al., 2014*; *Imlay and Odom, 2014*). This proposed function is consistent with its localization to the cytoplasm and other cellular foci outside the apicoplast and mitochondrion (*Gabriel et al., 2015a*).

We first considered the model that this FPPS/GGPPS might have an essential role in producing GGPP required for apicoplast biogenesis. A recent study, however, identified a specific inhibitor (MMV019313) of PF3D7_1128400 that is lethal to parasites but does not impact apicoplast biogenesis (*Gisselberg et al., 2018*). We independently confirmed that lethal treatment with MMV019313 did not affect apicoplast branching in the PfMev line (*Figure 5—figure supplement 2*). These observations strongly suggest that the cytosolic FPPS/GGPPS is not the origin of the PPS activity required for apicoplast biogenesis. Therefore, we turned our attention to the second isoprenoid synthase homolog in *P. falciparum*, PF3D7_0202700, which shares 23% sequence identity with avian FPPS.

Like the FPP/GGPP synthase, PF3D7_0202700 retains the DDXXD sequence dyad expected for a polyprenyl pyrophosphate synthase. In addition, the amino acid floor of PF3D7_0202700 features a sterically smaller Gly-Ser dyad upstream of the first DDXXD (*Figure 5A* and *Figure 5—figure*

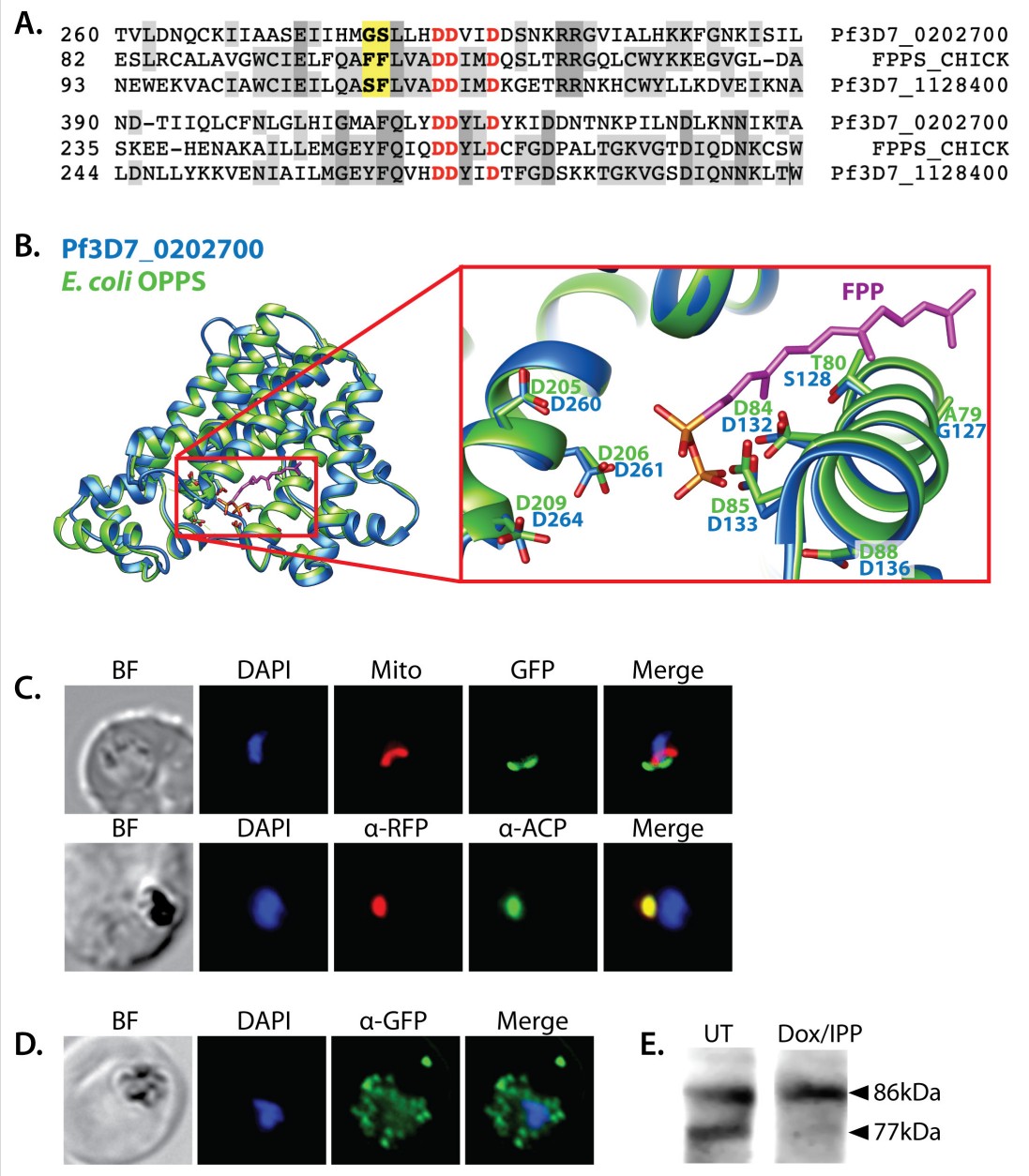

**Figure 5.** Sequence alignment and localization of the polyprenyl synthase (PPS) PF3D7_0202700 to the apicoplast. (**A**) Focal sequence alignment of avian farnesyl pyrophosphate synthase (FPPS, Uniprot P08836) with its two *Plasmodium falciparum* homologs reveals the presence of conserved metal-binding DDXXD motifs (red) expected for PPS activity and chain-length determination residues (yellow) upstream of the first DDXXD. (**B**) Homology model of PF3D7_0202700 using *Escherichia coli* octaprenyl pyrophosphate synthase (PDB 3WJK) as a structural template. The inset box is an enlargement of the active-site pocket showing the conserved Asp residues, bound FPP substrate, and product length-determining residues just upstream of the first DDXXD motif. (**C**) Bright-field (BF), fluorescence images (top) of live parasites episomally expressing PPS-GFP and stained with 10 nM Mitotracker Red and (bottom) immunofluorescence analysis (IFA) images of fixed parasites episomally expressing PPS-RFP stained with anti-RFP and anti-apicoplast ACP antibodies. (**D**) IFA images of fixed parasites expressing PPS-GFP that had been treated for >7 days with 2 µM doxycycline (Dox) and 200 µM isopentenyl pyrophosphate (IPP) (to stably induce apicoplast loss) and stained with anti-GFP antibody to visualize PPS distribution. (**E**) Western blot analysis of untreated (UT) or Dox/IPP-treated parasites episomally expressing PPS-RFP. PPS-RFP expression was visualized using an anti-RFP antibody. The full western blot image is included in *Figure 5—source data 1*.

The online version of this article includes the following source data and figure supplement(s) for figure 5:

**Source data 1.** Uncropped western blot image detecting polyprenyl synthase (PPS)-RFP expression in parasites.

**Figure supplement 1.** Full sequence alignment of PF3D7_0202700, PF3D7_1128400, and avian farnesyl pyrophosphate synthase (FPPS) (Uniprot P08836).

*Figure 5 continued on next page*

*Figure 5 continued*

**Figure supplement 2.** Epifluorescence microscopy images and statistical analysis of PfMev and D10 parasites treated with 10 µM MMV091313.

**Figure supplement 3.** Results of sequence-similarity searches for PF3D7_0202700 using NCBI BLAST and MPI HHpred.

**Figure supplement 4.** Additional epifluorescence microscopy images of Dd2 parasite episomally expressing polyprenyl synthase (PPS)-GFP or PPS-RFP.

*supplement 1*) that suggests an ability to synthesize longer-chain isoprenoids greater than four isoprene units. Consistent with these features, sequence similarity searches via NCBI BLAST (*Boratyn et al., 2013*) and MPI HHpred (*Zimmermann et al., 2018*) identify PPS homologs from bacteria, algae, and plants that share ~30% sequence identity with PF3D7_0202700 and have annotated functions in synthesizing polyprenyl isoprenoids of 4–10 units (*Figure 5—figure supplement 3*). Using *E. coli* octaprenyl pyrophosphate synthase (PDB 3WJK, 28% identity) as template, we generated a homology model of PF3D7_0202700 to visualize the possible structure of its active site (*Figure 5B*).

A prior in vitro study of PF3D7_0202700 function, using truncated recombinant protein expressed in *E. coli* or impure parasite extracts, reported an ability to synthesize polyprenyl-PP products of 8–11 isoprene units (*Tonhosolo et al., 2005*). Based on the authors' description, this truncated recombinant protein appears to have lacked one of the DDXXD motifs. Because of this difference from the native protein and the impurity of the parasite-derived protein, it remains possible that the native, pure protein has a distinct product spectrum than previously reported. Nevertheless, this in vitro activity and the general sequence features of PF3D7_0202700 support its function as a long-chain PPS.

Prior immunofluorescence studies of this PPS, using a polyclonal antibody raised against the truncated recombinant protein, were unable to localize PF3D7_0202700 to a specific sub-cellular compartment (*Tonhosolo et al., 2009*). Analysis of the protein sequence with MitoProt II or PlasmoAP suggested the presence of a subcellular-targeting leader sequence (*Foth et al., 2003*; *Claros and Vincens, 1996*), but ambiguous sequence features prevented a high-confidence targeting prediction to the mitochondrion or apicoplast. MitoProt II predicted a 55% probability of mitochondrial targeting. PlasmoAP strongly predicted an apicoplast-targeting transit peptide, but SignalP did not recognize the N-terminal sequence as a canonical signal peptide. To localize PPS within parasites, we engineered Dd2 *P. falciparum* lines to episomally express full-length PPS fused to either C-terminal GFP or RFP. In live parasites, focal PPS-GFP fluorescence was detected in a tubular compartment proximal to but distinct from the mitochondrion, as expected for apicoplast localization. Immunofluorescence analysis (IFA) of the PPS-RFP line revealed strong co-localization between PPS-RFP and the apicoplast ACP (*Figure 5C* and *Figure 5—figure supplement 4*).

To further confirm apicoplast targeting of PPS, we stably disrupted the apicoplast in the PPS-GFP Dd2 line by culturing these parasites in 2 µM doxycycline and 200 µM IPP for 1 week (*Yeh and DeRisi, 2011*; *Sigala et al., 2015*). As expected for an apicoplast-targeted protein, the PPS-GFP signal in these parasites displayed a constellation of dispersed fluorescent foci, rather than the concentrated signal observed in untreated parasites (*Figure 5D* and *Figure 5—figure supplement 4*). Western blot analysis of the PPS-RFP parasites revealed two bands at the expected sizes for precursor protein and a smaller, N-terminally processed mature form, as expected for import into the apicoplast (*Figure 5E*; *Waller et al., 2000*). In the apicoplast-disrupted parasites, however, only a single PPS-RFP band for the precursor protein was detected, consistent with loss of apicoplast import and lack of transit peptide removal (*Yeh and DeRisi, 2011*; *Dahl et al., 2006*). On the basis of these observations, we conclude that PF3D7_0202700 is an apicoplast-targeted PPS that is imported into the organelle and N-terminally processed to a mature form. This localization, the predicted ability of this enzyme to synthesize polyprenyl-PPs longer than four isoprene units, and our observation that decaprenol rescued FOS-induced defects in apicoplast biogenesis all suggested a critical role for this protein in apicoplast maintenance.

## PPS is essential for parasite viability and apicoplast biogenesis

The genomic locus for PF3D7_0202700 was reported to be refractory to disruption in recent genome-wide KO studies in *P. berghei* (*Bushell et al., 2017*) and *P. falciparum*, (*Zhang et al., 2018*) suggesting an essential function. To directly test its functional essentiality in *P. falciparum*, we used CRISPR/Cas9 (*Ghorbal et al., 2014*) to tag the endogenous gene in Dd2 parasites to encode a C-terminal hemagglutinin (HA)-FLAG epitope fusion and the aptamer/TetR-DOZI system (*Ganesan et al., 2016*) that

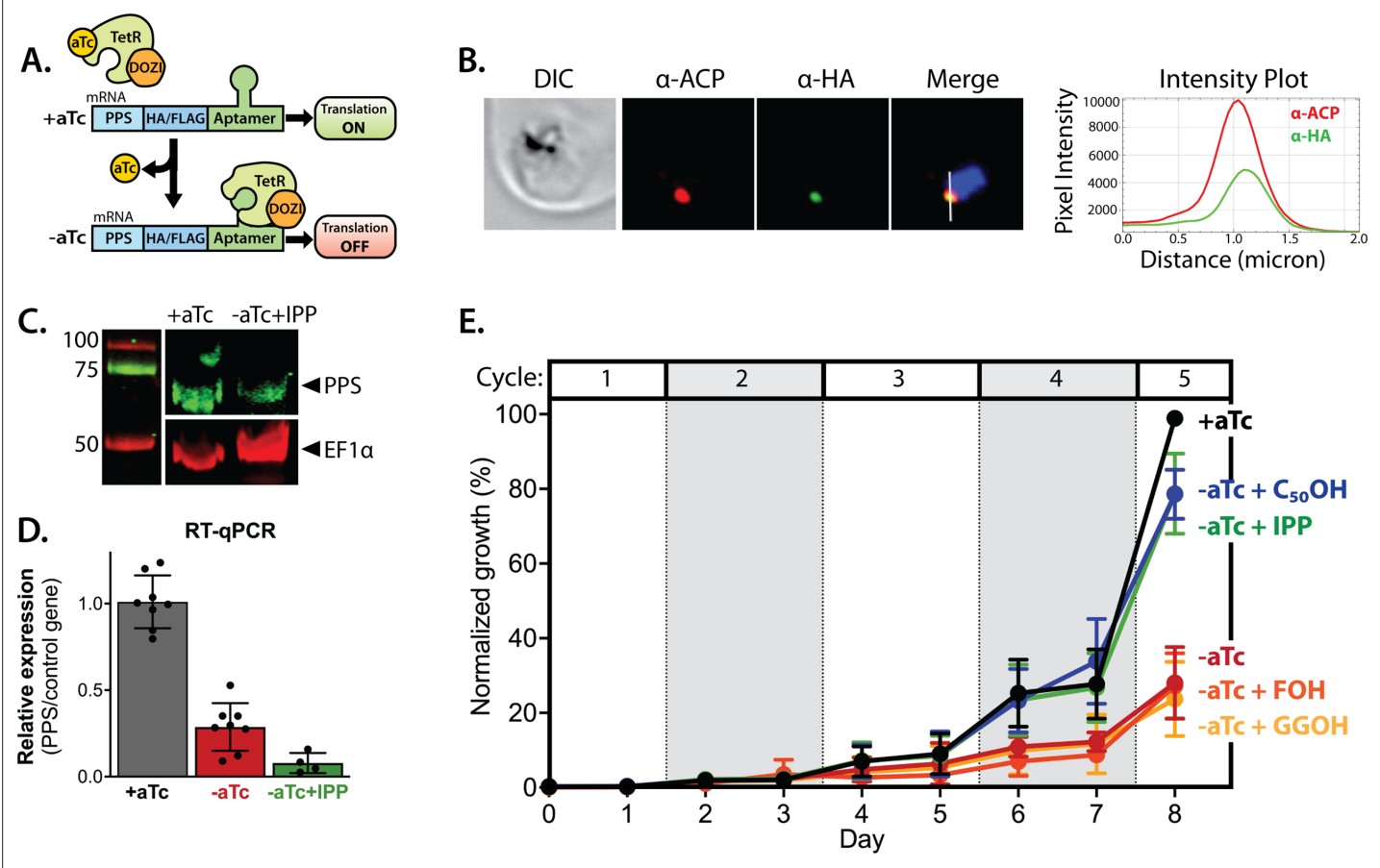

**Figure 6.** Polyprenyl synthase (PPS) (PF3D7_0202700) is essential for parasite viability and apicoplast function. (**A**) Schematic depiction of the aptamer/ TetR-DOZI system for ligand-dependent protein expression. (**B**) Immunofluorescence analysis of fixed parasites endogenously expressing PPS-HA-FLAG and stained with anti-acyl carrier protein (ACP) and anti-hemagglutinin (HA)-tag antibodies. The intensity plot displays the overlap in pixel intensity for ACP and HA signals as a function of distance along the white line in the merged image. (**C**) Western blot of endogenously tagged PPS-HA/FLAG showing detection of tagged PPS at the expected size for mature PPS of ~60 kDa for growth in +aTc conditions but diminished signal for parasites grown -aTc + isopentenyl pyrophosphate (IPP) for 5 days. Densitometry of the PPS signal relative to the EF1α loading control indicated a threefold signal reduction in +aTc versus -aTc/ + IPP conditions. (**D**) RT-qPCR analysis of PPS transcript levels (normalized to the average of two nuclear control genes) in biological replicate samples of synchronous parasites cultured for 72 hr ± aTc or -aTc/ + IPP. (**E**) Synchronous growth assay of Dd2 parasites tagged at the PPS locus with the aptamer/TetR-DOZI system and grown ±aTc and ± 200 µM IPP or 5 µM farnesol (FOH), geranylgeraniol (GGOH), or decaprenol (C$_{50}$-OH). Parasitemia values for each condition are the average ± SD of three biological replicates.

The online version of this article includes the following source data and figure supplement(s) for figure 6:

**Source data 1.** Uncropped western blot image detecting endogenous polyprenyl synthase (PPS)-hemagglutinin (HA)/FLAG in parasites.

**Source data 2.** Uncropped Southern blot image to probe editing of polyprenyl synthase (PPS) gene.

**Figure supplement 1.** Scheme for modification of the polyprenyl synthase (PPS) genomic locus to integrate the aptamer/TetR-DOZI system and Southern blot confirming correct integration.

**Figure supplement 2.** Additional immunofluorescence microscopy images showing co-localization of endogenous polyprenyl synthase (PPS) and apicoplast acyl carrier protein (ACP).

**Figure supplement 3.** Blood-smear images of Dd2 parasites tagged at the polyprenyl synthase (PPS) locus with the aptamer/TetR-DOZI system and grown ±aTc for 8 days.

enables ligand-dependent regulation of protein expression using the non-toxic small molecule, anhydrotetracycline (aTc). In this system, normal proteinexpression occurs +aTc and translational repression is induced upon aTc washout (*Figure 6A*).

We first confirmed correct integration into the genomic locus with the expected genotype in both polyclonal and clonal parasites by Southern blot (*Figure 6—figure supplement 1*). We then used immunofluorescence microscopy to confirm co-localization of the endogenous PPS with apicoplast

ACP (*Figure 6B* and *Figure 6—figure supplement 2*). Expression of the ~60 kDa HA-FLAG-tagged, endogenous mature protein was detected by anti-HA-tag western blot in +aTc conditions, and its signal relative to the loading control was reduced nearly threefold in parasites grown in -aTc + IPP conditions (*Figure 6C*). Our detection of only the mature form of endogenous PPS contrasts with our detection of both the precursor and mature forms of episomally expressed PPS-RFP (*Figure 5E*), suggesting that the precursor form may preferentially accumulate when PPS is over-expressed. We also quantified PPS transcript levels by RT-qPCR and observed robust KD of PPS mRNA levels by the second intraerythrocytic cycle of parasite growth in -aTc conditions (*Figure 6D*). The fate of target mRNA in the aptamer/TetR-DOZI system has not been characterized in depth. Our data are consistent with a prior report (*Maruthi et al., 2020*) and suggest that TetR-DOZI binding after aTc washout leads to mRNA transcript degradation, possibly within stress granules targeted by DOZI-bound transcripts (*Ganesan et al., 2016*).

To test PPS essentiality for blood-stage parasite growth, we synchronized PPS KD parasites and monitored their growth ±aTc over multiple intraerythrocytic lifecycles. In the presence of aTc, culture parasitemia expanded in a step-wise fashion over the ~10 days of the growth assay such that the culture needed to be split multiple times to avoid over-growth (*Figure 6E*). Without aTc, however, the culture grew normally over the first three intraerythrocytic cycles but showed a major growth defect in the fourth cycle consistent with extensive parasite death observed by blood smear (*Figure 6E* and *Figure 6—figure supplement 3*). Parasite growth under -aTc conditions was strongly rescued in the presence of 200 µM exogenous IPP (*Figure 6E*), indicating an essential PPS function specific to the apicoplast.

To test a role for PPS in synthesizing polyprenyl-PP groups, we attempted to rescue parasite growth in -aTc conditions by adding 5 µM FOH, GGOH, or decaprenol. We observed that only decaprenol, but not FOH or GGOH, rescued parasite growth in -aTc conditions, and the magnitude of rescue by decaprenol was comparable to IPP (*Figure 6E*). These observations strongly suggest that PPS has an essential function downstream of IPP synthesis in converting isoprenoid precursors into longer-chain linear polyprenyl-PPs containing at least 5–10 isoprene units.

To test if PPS function is required for apicoplast biogenesis, we cultured PPS KD parasites in +aTc or -aTc/ + IPP conditions for 12 days and then assessed apicoplast morphology in fixed parasites by αACP immunofluorescence. We reasoned that IPP would rescue parasite viability upon PPS KD but not interfere with assessing any defect in apicoplast biogenesis if IPP synthesis were upstream of PPS function. IFA revealed that ~30% of parasites cultured in -aTc conditions had a dispersed ACP signal indicative of apicoplast disruption (*Figure 7—figure supplement 1*). Although this observation supports a critical role for PPS in apicoplast biogenesis, we wondered why PPS KD did not result in a higher fraction of parasites with disrupted apicoplast. We hypothesized that residual PPS expression resulting from incomplete KD combined with high IPP levels due to culture supplementation and endogenous MEP pathway activity might enable sufficient synthesis of polyprenols to attenuate the impact of PPS KD on apicoplast biogenesis.

To test this hypothesis and the contribution of MEP pathway activity to the observed phenotype, we synchronized parasites to the ring stage and cultured them in ±aTc conditions for 96 hr (two 48 hr growth cycles) to knock down PPS expression before adding FOS and IPP at the start of the third growth cycle (*Figure 7A*). In this experiment, FOS was expected to inhibit endogenous MEP pathway activity without impacting apicoplast biogenesis since it was added concurrently with IPP, which fully rescues parasites from growth and apicoplast defects induced by FOS (*Figure 1*; *Yeh and DeRisi, 2011*; *Gisselberg et al., 2013*). We first used IFA to assess apicoplast morphology in schizonts at the end of the third growth cycle (38 hr after adding FOS and IPP). We observed normal apicoplast elongation in +aTc parasites but focal, unbranched apicoplast morphology in the vast majority (>80%) of -aTc parasites (*Figure 7B and C*, and *Figure 7—figure supplement 2*). Substitution of IPP with FOH or GGOH resulted in a nearly identical apicoplast elongation defect in -aTc parasites. In contrast, substituting IPP with decaprenol resulted in normal apicoplast elongation in both +aTc and -aTc parasites (*Figure 7B and C*, and *Figure 7—figure supplement 2*). The selective ability of decaprenol to rescue apicoplast-branching defects in -aTc conditions strongly supports an essential role for PPS in synthesizing long-chain polyprenyl isoprenoids required for apicoplast biogenesis.

To further test this conclusion, we maintained parasites in ±aTc conditions with FOS and IPP for two additional growth cycles (total of five 48 hr cycles, *Figure 7A*). Parasites cultured +aTc displayed

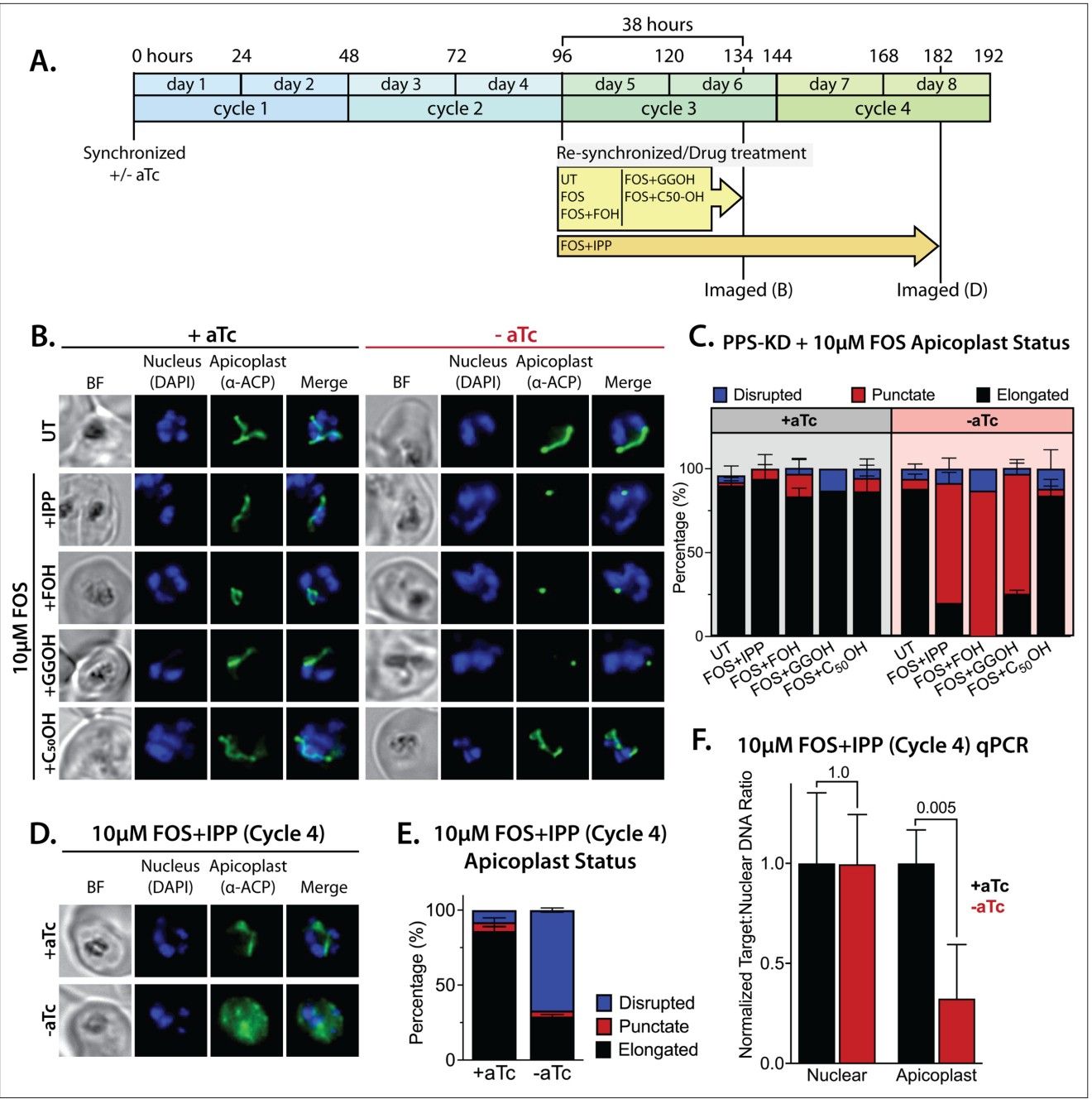

**Figure 7.** Polyprenyl synthase (PPS) is required for apicoplast biogenesis. (**A**) Scheme summarizing growth of synchronized PPS knockdown parasites cultured ±aTc, re-synchronized and treated with 10 µM fosmidomycin (FOS) ±200 µM isopentenyl pyrophosphate (IPP) or 5 µM farnesol (FOH), geranylgeraniol (GGOH), or decaprenol (C50-OH) at 96 hr after initial synchronization, and imaged at 134 and 182 hr after initial synchronization. (**B**) Immunofluorescence analysis (IFA) of PPS knockdown parasites cultured as described in panel A and imaged at 134 hr (day 6) after initial synchronization to assess apicoplast morphology ±aTc. (**C**) Population analysis of apicoplast morphology for 50 total parasites imaged for each condition in panel A from two independent experiments. Apicoplast morphologies were scored as punctate (focal), elongated, or disrupted (dispersed); counted; and plotted by histogram as the fractional population with the indicated morphology (UT = untreated). Error bars represent standard deviations from replicate experiments. (**D**) IFA of PPS knockdown parasites cultured ±aTc + FOS + IPP and imaged at 182 hr (day 8) after initial synchronization to assess apicoplast morphology ±aTc. (**E**) Population analysis of apicoplast morphology for 50 total parasites imaged for each condition in panel D and analyzed as in panel C. (**F**) Quantitative PCR analysis of the apicoplast:nuclear (Api:Nu) genome ratio for parasites cultured ±aTc and imaged in panel D, based on amplification of apicoplast TufA (PF3D7_API02900) or nuclear ADSL (PF3D7_0206700) relative to nuclear I5P (PF3D7_0802500) genes. Indicated qPCR ratios were normalized to +aTc in each case and are the average ± SD of three biological replicates. Significance of ±aTc differences was analyzed by two-tailed unpaired t-test to determine the stated p value. All parasite samples collected for IFA were imaged by bright-field (BF) and epifluorescence microscopy, with visualization of parasite nuclei by DAPI staining and apicoplast by an anti-apicoplast acyl carrier protein (ACP) antibody.

*Figure 7 continued on next page*

Figure 7 continued

The online version of this article includes the following source data and figure supplement(s) for figure 7:

**Source data 1.** Absolute parasite counts for microscopy experiments.

**Figure supplement 1.** Immunofluorescence analysis (IFA) images and analysis of apicoplast morphology in polyprenyl synthase (PPS) knockdown parasites grown +aTc or -aTc/ + isopentenyl pyrophosphate (IPP) (200 µM) for 12 days.

**Figure supplement 2.** Additional immunofluorescence analysis (IFA) images of polyprenyl synthase (PPS) knockdown parasites treated as in *Figure 7B*.

**Figure supplement 3.** Additional immunofluorescence analysis (IFA) images of polyprenyl synthase (PPS) knockdown parasites treated as in *Figure 7D*.

normal elongated apicoplast morphology. In contrast, the -aTc (+FOS and IPP) culture predominantly contained parasites with a dispersed ACP signal indicative of apicoplast loss (*Figure 7D and E*, and *Figure 7—figure supplement 3*). These -aTc parasites also contained a strongly reduced qPCR signal for apicoplast genomic DNA, relative to +aTc parasites (*Figure 7F*). These results indicate that PPS is essential for apicoplast maintenance and inheritance by daughter parasites such that loss of PPS function (with IPP supplementation) results in parasite progeny lacking the intact organelle. This essential PPS function downstream of IPP synthesis by the MEP pathway is sufficient to explain our observation that blocking pathway activity by FOS or ΔDXS inhibits apicoplast biogenesis (*Figure 1*).

## No evidence for PPS function in carotenoid synthesis

Despite its strong sequence similarity to known PPSs that catalyze the head-to-tail condensation of isoprenoid precursors, PF3D7_0202700 has also been proposed to catalyze the biochemically distinct head-to-head condensation of 20-carbon GGPP groups into 40-carbon phytoene and thus function as a phytoene synthase (PSY) within a broader pathway of carotenoid biosynthesis proposed to exist in *Plasmodium* parasites (*Figure 8A*; *Tonhosolo et al., 2009*; *Gabriel et al., 2015b*). PPSs and PSYs are mechanistically distinct enzymes that lack significant sequence similarity but are thought to share a common isoprenoid-related protein fold that reflects their ancient divergence from a common ancestral enzyme (*Thulasiram and Poulter, 2006*; *Bouvier et al., 2005*). Given the mechanistic differences between head-to-tail and head-to-head condensation of isoprenoids (*Figure 8A*), which involve distinct positioning of substrate pyrophosphate groups within each active site, there is no known enzyme that is capable of catalyzing both reactions (*Bouvier et al., 2005*). Thus, the proposal of dual PPS and PSY functions for PF3D7_0202700 is without biochemical precedent. Nevertheless, we considered whether this protein might also have PSY function and evaluated whether existing observations supported or contradicted a proposed role for this protein in carotenoid biosynthesis.

As noted previously, untargeted sequence similarity searches via NCBI BLAST (*Boratyn et al., 2013*) and MPI HHpred (*Zimmermann et al., 2018*) with PF3D7_0202700 as the query sequence only identify PPS homologs from bacteria, algae, and plants (*Figure 5—figure supplement 3*) and fail to identify PSY homologs. Furthermore, targeted pairwise alignments show no evidence of significant sequence homology between PF3D7_0202700 and confirmed eukaryotic or prokaryotic PSY sequences from *Arabidopsis thaliana* (Uniprot P37271, chloroplast-targeted) (*Zhou et al., 2015*) or *Erwinia herbicola* (*Pantoea agglomerans*, Uniprot D5KXJ0) (*Iwata-Reuyl et al., 2003*), respectively. Finally, the prior proposal of PSY activity by PF3D7_0202700 was based in part on its sequence similarity to an annotated PSY from *Rubrivivax gelatinosus* bacteria (NCBI accession BAA94032) that also appeared to contain sequence features expected of a head-to-tail PPS (*Tonhosolo et al., 2009*). We noted that the functional annotation of this bacterial protein was subsequently revised to a GGPPS (Uniprot I0HUM5) (*Nagashima et al., 2012*), thus explaining its sequence similarity to PF3D7_0202700 and the homology of both proteins to known PPSs. On the basis of these sequence analyses, we considered it unlikely that PF3D7_0202700 had dual activity as a PSY.

The prior work studied the antiparasitic effects of the squalene synthase inhibitor, zaragozic acid (ZA, also called squalestatin), that inhibited blood-stage *P. falciparum* growth (EC$_{50}$ ~5 µM) and was proposed to specifically target PF3D7_0202700 based on observation of an approximately sixfold increase in EC$_{50}$ for parasites episomally expressing a second copy of this protein (*Gabriel et al., 2015b*). We repeated these experiments in Dd2 parasites and observed a similar EC$_{50}$ of ~10 µM for ZA that increased fivefold to ~50 µM in Dd2 parasites episomally expressing PPS-RFP (*Figure 8— figure supplement 1*). However, in contrast to PPS KD (*Figure 6D*), lethal growth inhibition by ZA was not rescued by exogenous IPP (*Figure 8—figure supplement 1*) and did not affect apicoplast

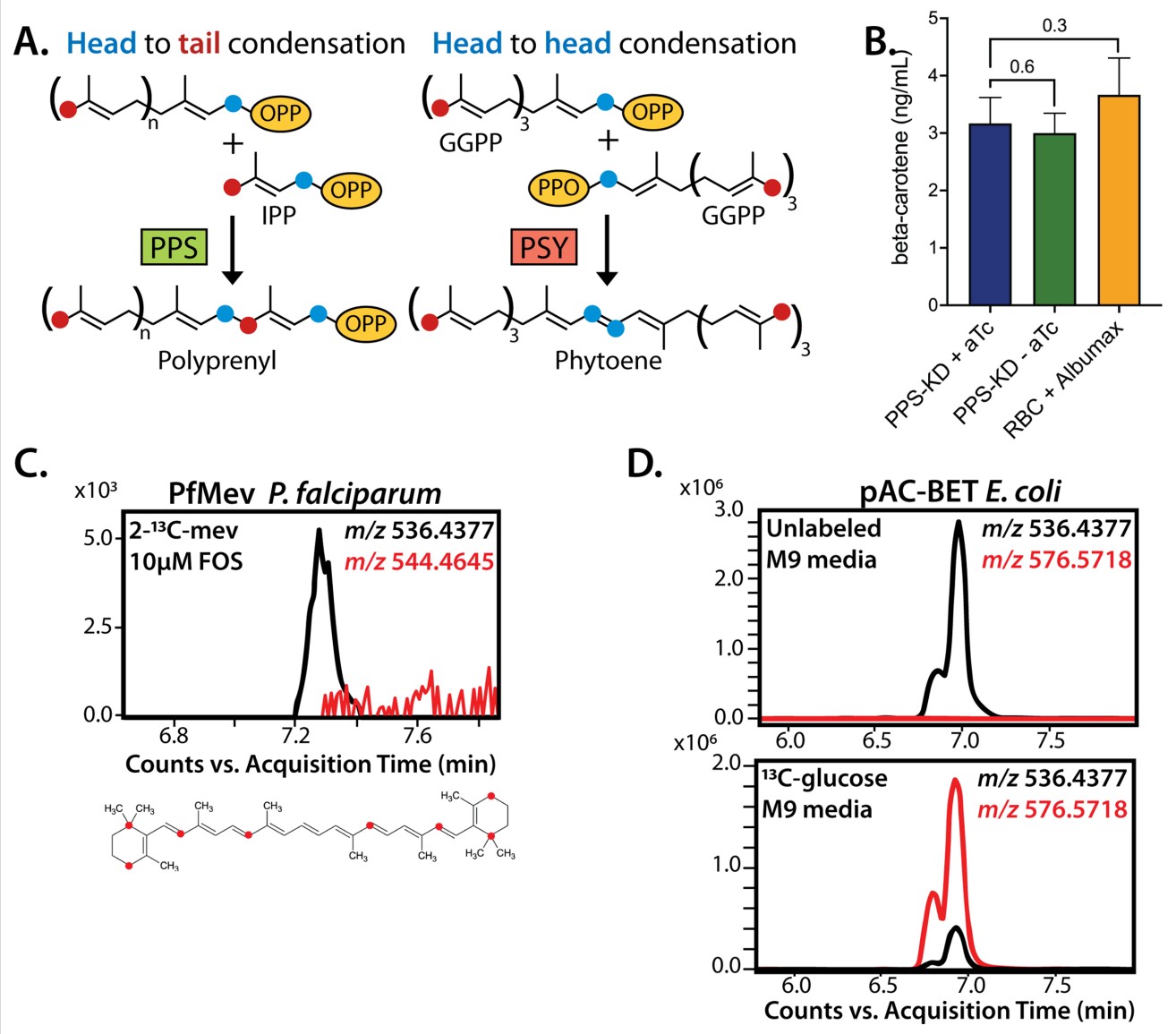

**Figure 8.** No evidence that polyprenyl synthase (PPS) contributes to carotenoid synthesis by *Plasmodium falciparum*. (**A**) Schematic depiction of head-to-tail arrangement of prenyl groups during polyprenyl-PP synthesis versus head-to-head arrangement of geranylgeranyl-PP groups during phytoene synthesis. (**B**) Mass spectrometry determination of unlabeled β-carotene levels in PPS knockdown parasites grown for 6 days ± aTc or in uninfected red blood cells incubated in complete media containing Albumax. Measured β-carotene levels are the average ± SD of three biological replicates, whose differences were analyzed by two-tailed unpaired t-test for significance (p values given relative to +aTc sample). (**C**) Intensity versus retention time plot for liquid chromatography-mass spectrometry determination of unlabeled and $^{13}$C-labeled β-carotene in NF54 PfMev parasites cultured for 6 days in 50 µM 2-$^{13}$C-mevalonate and 10 µM fosmidomycin. Below: schematic depiction of the eight carbon atoms in β-carotene expected to be labeled with $^{13}$C for synthesis from isopentenyl pyrophosphate (IPP) derived from 2-$^{13}$C-mevalonate in PfMev parasites. (**D**) Intensity versus retention time plot for liquid chromatography-mass spectrometry determination of unlabeled and $^{13}$C-labeled β-carotene in pAC-BETAipi *Escherichia coli* grown in unlabeled or fully $^{13}$C-labeled glucose as the sole carbon source in M9 minimal media. The two peaks reflect the presence of an isomeric mix of all-trans and cis β-carotene produced by the pAC-BETAipi *E. coli*, as previously reported (**Cunningham and Gantt, 2005**).

The online version of this article includes the following figure supplement(s) for figure 8:

**Figure supplement 1.** Forty-eight hour growth inhibition curves for treatment of Dd2 parasites with zaragozic acid without or with episomal expression of polyprenyl synthase (PPS)-RFP or 200 µM isopentenyl pyrophosphate (IPP).

**Figure supplement 2.** Epifluorescence microscopy images of D10 parasites treated with 160 µM zaragozic acid as synchronized rings and imaged for ACP$_L$-GFP and Hoescht 36 hr later as multinuclear schizonts.

**Figure supplement 3.** Fragment ion spectrum for unlabeled β-carotene determined by tandem mass spectrometry of β-carotene commercial standard.

**Figure supplement 4.** Intensity versus retention time plot for liquid chromatography-mass spectrometry determination of unlabeled β-carotene in Albumax I.

elongation (*Figure 8—figure supplement 2*). These contrasting phenotypes strongly suggest that PPS, and more broadly the apicoplast, are not uniquely targeted by ZA. The basis for why PPS over-expression reduces parasite sensitivity to ZA is unclear but may reflect drug interactions with broader isoprenoid metabolism outside the apicoplast that are rescued, directly or indirectly, by reaction products of PPS.

β-Carotene, a 40-carbon carotenoid derived from phytoene, was previously detected by MS in extracts of *P. falciparum*-infected erythrocytes and suggested to be biosynthesized by parasites based on the lack of detection in extracts of uninfected erythrocytes (*Tonhosolo et al., 2009*). Using our PPS KD line, we tested whether translational repression of PF3D7_0202700 impacted detectable levels of β-carotene in parasites, as predicted to occur if PPS also functioned as a PSY. After synchronization, PPS KD parasites were grown ±aTc for 120 hr and harvested at the end of the third intraerythrocytic growth cycle, which immediately precedes the growth defect observed in *Figure 6E*. Saponin pellets of these parasites were extracted in acetone and analyzed by liquid chromatography/tandem mass spectrometry (LC-MS/MS) for β-carotene (*Figure 8—figure supplement 3*). We observed indistinguishable low levels of β-carotene in both samples (*Figure 8B*), providing no evidence that PPS plays a role in carotenoid biosynthesis.

Although uninfected erythrocytes washed in AlbuMAX-free RPMI lacked detectable β-carotene, we observed that extracts of uninfected erythrocytes incubated in complete RPMI medium containing AlbuMAX I had β-carotene levels that were nearly identical to extracts of parasite-infected erythrocytes (*Figure 8B*). Analysis of AlbuMAX (Thermo Fisher catalog #11020021) by MS revealed modest levels of β-carotene (*Figure 8—figure supplement 4*), consistent with the bovine origin of AlbuMAX (lipid-rich bovine serum albumin) and the plant-based diet of these animals expected to contain β-carotene. These results are sufficient to explain the presence of carotenoids like β-carotene in parasite extracts. In summary, we find no evidence that PPS function contributes to β-carotene levels in *P. falciparum*-infected erythrocytes, which we suggest non-specifically take up exogenous plant-derived β-carotene associated with AlbuMAX in the culture medium.

Parasite-infected erythrocytes were previously reported to incorporate $^{3}$H-labeled GGPP into biosynthetic products that had reverse-phase HPLC retention times similar to all-trans-lutein or β-carotene standards, suggesting de novo synthesis of these isoprenoid products (*Tonhosolo et al., 2009*). Because the extracted products were radioactive, their identity could not be directly confirmed by MS/MS. To directly test if blood-stage *P. falciparum* parasites incorporate isoprenoid precursors into β-carotene, as predicted for active biosynthesis, we cultured the PfMev parasites in 50 μM of 2-$^{13}$C-mevalonate in the presence of 10 μM FOS. This strategy was chosen to inhibit MEP pathway activity, ensure full $^{13}$C-labeling of the endogenous IPP and DMAPP precursor pool produced by the cytoplasmic bypass enzymes, and result in a distinguishable 8 Da mass increase for any β-carotene derived from de novo synthesis. We previously showed that this strategy results in complete $^{13}$C-labeling of endogenous IPP and FPP (*Swift et al., 2020b*). Parasites were expanded to high parasitemia over several days under $^{13}$C-labeling conditions before extraction and analysis by MS. Although we readily detected unlabeled β-carotene (*m/z* 536.4), which we attribute to culture medium AlbuMAX, we were unable to detect $^{13}$C-labeled β-carotene (*m/z* 544.4) (*Figure 8C*). In contrast to the parasite analysis, we readily detected fully $^{13}$C-labeled β-carotene (*m/z* 576.6) produced by *E. coli* bacteria engineered to biosynthesize β-carotene (*Cunningham and Gantt, 2005*) and grown in minimal M9 medium with uniformly labeled $^{13}$C-glucose as the sole carbon source (*Figure 8D*). In summary, we find no evidence for PSY function by PF3D7_0202700 or for de novo carotenoid synthesis by blood-stage *P. falciparum* parasites. Collectively, our data strongly support the conclusion that PF3D7_0202700 functions exclusively as a PPS required for apicoplast biogenesis.

## Discussion

Biosynthesis of the isoprenoid precursors, IPP and DMAPP, is a well-established essential function of the *Plasmodium* apicoplast, but prior work has focused nearly exclusively on the critical roles of isoprenoids for diverse cellular processes outside this organelle (*Guggisberg et al., 2014*; *Yeh and DeRisi, 2011*; *Kennedy et al., 2019a*; *Imlay and Odom, 2014*). We have elucidated a novel arm of isoprenoid metabolism within the apicoplast that is required for biogenesis of this critical organelle (*Figure 9*). This discovery expands the paradigm for isoprenoid utilization by malaria parasites,

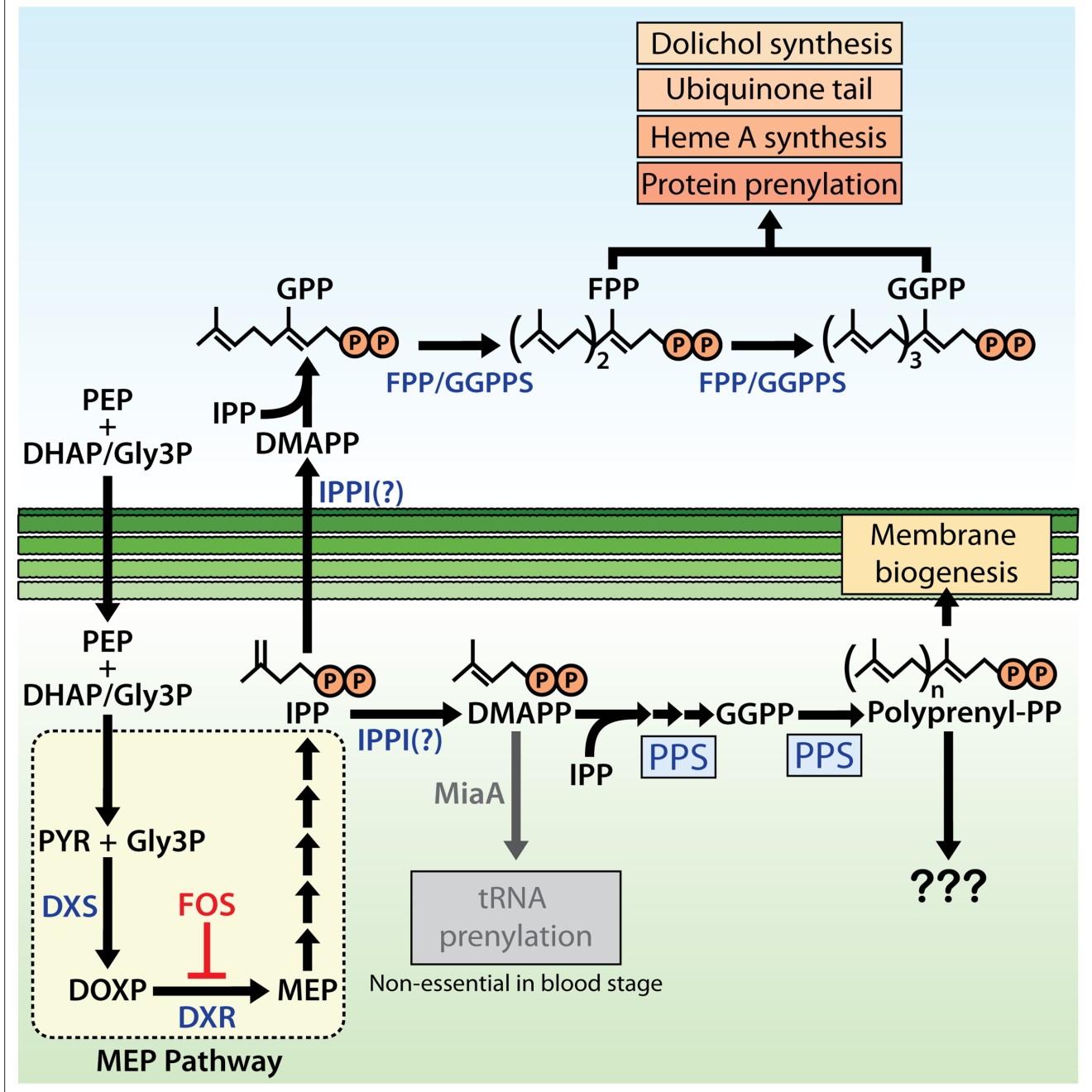

**Figure 9.** Schematic diagram of apicoplast isoprenoid metabolism in blood-stage *Plasmodium falciparum* parasites. IPPI = isopentenyl pyrophosphate (IPP) isomerase, PEP = phosphoenolpyruvate, DHAP = dihydroxyacetone phosphate, PYR = pyruvate, Gly3P = glyceraldehyde-3-phosphate. Question marks indicate uncertainty in the identity of the proposed IPP isomerase and in the role of polyprenyl isoprenoid products of polyprenyl synthase (PPS) in apicoplast biogenesis. For simplicity, we have depicted PPS targeting to the apicoplast matrix. However, further experiments will be needed to test and specify sub-organellar targeting of PPS to the apicoplast matrix and/or intermembrane spaces.

uncovers a novel essential feature of apicoplast biology, and identifies a key enzyme in this pathway suitable for development as a therapeutic target.

## Implications for general understanding of apicoplast functions

Our study, which was inspired by prior hints in the literature (*Nair et al., 2011*; *Bowman et al., 2014*; *Goodman and McFadden, 2014*), firmly establishes a novel essential role for MEP pathway activity in supporting apicoplast biogenesis, in addition to its recognized role producing IPP required outside

this organelle. Prior work has clearly established that apicoplast maintenance pathways are required to support IPP synthesis and export for essential cellular use (*Yeh and DeRisi, 2011*; *Kennedy et al., 2019a*; *Gisselberg et al., 2013*; *Dahl et al., 2006*), but the reverse dependence of apicoplast biogenesis on IPP was not previously recognized. Thus, IPP synthesis by the MEP pathway requires apicoplast maintenance, which in turn depends on IPP synthesis. The two processes are convoluted and interdependent. This reciprocal dependence between organelle maintenance and metabolic outputs may extend to other apicoplast pathways in mosquito- and liver-stage *P. falciparum* as well as *Toxoplasma gondii*, where additional apicoplast outputs, such as fatty acids, contribute to parasite fitness and may support apicoplast maintenance (*Yu et al., 2008*; *Shears et al., 2015*; *van Schaijk et al., 2014*; *Krishnan et al., 2020*; *Mazumdar et al., 2006*).

Our results also support the emerging paradigm (*Uddin et al., 2018*; *Amberg-Johnson et al., 2017*; *Boucher and Yeh, 2019*; *Okada et al., 2020*) that inhibition of apicoplast maintenance pathways can kill parasites with first-cycle kinetics that defy the delayed-death phenotype commonly observed for translation-blocking antibiotics such as doxycycline that target organelle housekeeping (*Dahl and Rosenthal, 2007*; *Uddin et al., 2018*). Indeed, blocking IPP synthesis causes same-cycle defects in apicoplast biogenesis, which are expected to produce non-viable parasite progeny independent of lethal dysfunctions in isoprenoid-dependent metabolism outside the organelle. Analysis of the timing of FOS-induced defects in apicoplast branching also provides an unexpected and incisive window into the differential compartmentalization of IPP essentiality in parasites. We observed that FOS-treated parasites display apicoplast-elongation defects in early schizogony but continue to divide nuclear DNA and transition into mature schizonts before stalling prior to segmentation (*Figure 1A*). Thus, the critical role of IPP for apicoplast biogenesis precedes the broader cellular need for IPP outside the organelle in mature schizonts, suggested by recent works to predominantly reflect essential roles for IPP-dependent protein prenylation (*Kennedy et al., 2019a*; *Howe et al., 2013*). Although MEP pathway activity begins in ring-stage parasites (*Zhang et al., 2011*; *Cassera et al., 2004*), we observed identical inhibition of apicoplast elongation in schizonts independent of whether FOS was added to rings concomitant with synchronization or to trophozoites 12 hr after synchronization (*Figure 1A*). This observation suggests that IPP utilization in the apicoplast depends on de novo synthesis rather than a pre-existing metabolite pool, possibly because IPP does not accumulate in the apicoplast and/or that IPP synthesis within the organelle is differentially partitioned for export and internal utilization.

## Localization of PPS to the apicoplast

Our studies identified PF3D7_0202700 as an apicoplast-targeted PPS based on its co-localization with apicoplast ACP, dispersed localization upon apicoplast disruption, and N-terminal processing in an apicoplast-dependent manner. Nevertheless, the sequence features that target PPS to the apicoplast are somewhat obscure. Imported apicoplast proteins canonically contain N-terminal signal and transit peptides that are proteolytically removed in the endoplasmic reticulum (ER) and apicoplast, respectively (*Waller et al., 2000*; *van Dooren et al., 2002*). The N-terminus of PPS contains a transit peptide recognized by PlasmoAP (*Foth et al., 2003*) but is not recognized by SignalP (*Almagro Armenteros et al., 2019*) to contain a canonical signal peptide. This targeting ambiguity is not unique to PPS. We are aware of multiple apicoplast-targeted and imported proteins, including several identified in the prior proximity-biotinylation study of the apicoplast proteome (*Boucher et al., 2018*), that contain a recognizable transit peptide but lack an identifiable signal peptide by SignalP. These proteins include the key MEP pathway enzymes DXR (PF3D7_1467300) and IspD (PF3D7_0106900), holo ACP synthase (PF3D7_0420200), FabB/F (PF3D7_0626300), and the E1 subunit of pyruvate dehydrogenase (PF3D7_1446400). These observations suggest that protein N-termini in *Plasmodium* can have sequence properties compatible with apicoplast targeting that are broader and more heterogenous than ER-targeting sequences from other eukaryotes that comprise the training sets upon which SignalP is currently based. It remains a future challenge to understand these N-terminal properties and their relation to apicoplast targeting and import in *Plasmodium*.

N-terminal processing of apicoplast-targeted proteins is an exclusive property of known proteins imported into the organelle. Although multiple proteins have been shown or suggested to associate with the apicoplast exterior or target the outer apicoplast membrane, none of these proteins is N-terminally processed (e.g., ATG8, oTPT, and FTSH1) (*Amberg-Johnson et al., 2017*; *Mullin et al., 2006*;

*Tomlins et al., 2013*). N-terminal processing of PPS is therefore strongly suggestive of import into the apicoplast. Such processing, however, does not specify the sub-compartment within the apicoplast to which PPS is localized, which could include the matrix and/or multiple membranes or intermembrane spaces. Indeed, N-terminal processing has been reported for both matrix and membrane-targeted proteins imported into the apicoplast (*Mullin et al., 2006*). The central conclusions of our study are not affected by this ambiguity. For simplicity, we have depicted PPS localization within the apicoplast matrix (*Figure 9*), but more studies and higher resolution (e.g., electron microscopy with immunogold labeling) will be required to specify sub-compartmental targeting of PPS within the apicoplast. PPS lacks obvious membrane-targeting features but required overnight extraction in 2% SDS or LDS for detection by western blot. These features, together with low expression, may explain why PPS has not been detected in prior studies of the apicoplast proteome that used milder extraction conditions.

## Implications for other apicomplexan parasites

PPSs are diverse enzymes that perform a variety of cellular functions, whose specific roles can differ between organisms. The reliance of apicoplast biogenesis on isoprenoid synthesis in *Plasmodium* may differ from other apicomplexan parasites. *T. gondii* appears to express two PPS homologs, TGME49_224490 and TGME49_269430, that are ~30% identical (in homologous regions) to PF3D7_1128400 (FPPS/GGPPS) and PF3D7_0202700 (PPS), respectively. TGME49_224490 appears to be targeted to the mitochondrion in *T. gondii* (based on MitoProt and HyperLOPIT analyses) (*Claros and Vincens, 1996*; *Barylyuk et al., 2020*), in contrast to FPPS/GGPPS, its closest *P. falciparum* homolog, which localizes to the cytoplasm and other cellular foci outside the mitochondrion. TGME49_269430 does not appear to target the apicoplast in *T. gondii* (based on HyperLOPIT data), which contrasts with our determination of apicoplast targeting for PPS, the closest *P. falciparum* homolog. These observations suggest distinct cellular roles for enzyme homologs in the two parasites. Furthermore, a recent study (*Henkel et al., 2022*) showed that loss of MEP pathway activity in *T. gondii* (due to loss of apicoplast ferredoxin) does not impact apicoplast biogenesis, in contrast to our observations in *P. falciparum* based on FOS treatment, DXS deletion, and PPS KD. Finally, *Cryptosporidium parvum* lacks an apicoplast but retains two PPS homologs, CPATCC_003578 and CPATCC_001801, whose cellular targeting has not been reported. These differing enzyme localizations and phenotypes suggest differences in cellular isoprenoid utilization and metabolism between *T. gondii*, *P. falciparum*, and broader apicomplexan organisms that remain to be understood.

## Why does *Plasmodium* apicoplast biogenesis depend on IPP synthesis?

The essential function of PPS in apicoplast maintenance is sufficient to explain the apicoplast reliance on IPP synthesis unveiled by FOS treatment of parasites. Although the dominant polyprenyl-PP product of apicoplast PPS in parasites remains uncertain, sequence features, prior in vitro enzymology, and the ability of exogenous decaprenol but not GGOH or FOH to rescue PPS KD indicate that linear polyprenyl-PP products longer than 4 and as long as 10 isoprene units are critical for apicoplast maintenance (*Tonhosolo et al., 2005*). Prior work suggested a dual function for PF3D7_0202700 as a PSY that also condenses isoprenoid precursors (*Tonhosolo et al., 2009*; *Gabriel et al., 2015b*), but we found no evidence to support this proposed PSY function or carotenoid biosynthesis more broadly. Synthesis of octaprenyl-PP by PF3D7_0202700 was previously proposed to be critical for ubiquinone biosynthesis in the parasite mitochondrion (*Tonhosolo et al., 2005*). Localization of this protein to the apicoplast (*Figures 5 and 6*) and observation that exogenous IPP rescues the growth defects of its KD (*Figure 6E*) strongly suggest that its activity is not required for mitochondrial ubiquinone biosynthesis and that its essential function is specific to the apicoplast.

Plant chloroplasts synthesize linear polyprenyl isoprenoids to serve a wide variety of functions that are only partially understood but include key roles in light harvesting and photosynthesis, oxidative stress protection, and as precursors of signaling and defense molecules that function outside the chloroplast (e.g., abscisic acid, gibberellins, and terpenes) (*Akhtar et al., 2017*; *Joyard et al., 2009*; *Van Gelder et al., 2018*). The *Plasmodium* apicoplast has lost photosynthesis capabilities and has uncertain carotenoid and terpene synthesis capacity. Volatile terpenes (*Kelly et al., 2015*) and carotenoids (*Tonhosolo et al., 2009*) have been detected in *P. falciparum*-infected erythrocytes, but the parasite genome lacks enzyme homologs of the relevant synthases required for terpene and carotenoid biosynthesis (*Ralph et al., 2004*; *Guggisberg et al., 2014*). Furthermore, we found no evidence of

de novo β-carotene synthesis by parasites, and results herein as well as recent studies (*Emami et al., 2017*; *Miller and Odom John, 2020*) indicate that these metabolites can derive from erythrocyte and/or culture medium sources rather than parasite-specific synthesis. Thus, these known functions in chloroplasts seem uncertain or unlikely to explain apicoplast reliance on longer-chain PPS activity in malaria parasites. Longer-chain polyprenyl-PPs and related dolichols serve as membrane-bound glycan carriers for protein glycosylation, but these activities in *Plasmodium* appear to occur in the ER as they do in other organisms (*Imlay and Odom, 2014*; *Couto et al., 1999*; *Zimbres et al., 2020*).

Linear polyprenyl alcohols have been found to be important components of plant membranes, especially chloroplast membranes, where they are proposed to modulate membrane structure, fluidity, and dynamics (*Akhtar et al., 2017*; *Van Gelder et al., 2018*; *Hartley and Imperiali, 2012*; *Swiezewska and Danikiewicz, 2005*). In the absence of other known roles for longer-chain polyprenyl-PPs in the apicoplast, we hypothesize that linear polyprenols or polyprenyl phosphates may serve as critical components of the apicoplast membranes and be required for maintaining membrane fluidity during organelle biogenesis (*Figure 9*). A prior MS-based lipidomics study of isolated apicoplasts focused primarily on the fatty acid and phospholipid composition of this organelle and did not characterize isoprenoid components (*Botté et al., 2013*). Selective isotopic labeling of parasite-synthesized isoprenoids by 2-$^{13}$C-mevalonate in the PfMev line, combined with apicoplast isolation and our PPS KD line, can potentially identify specific apicoplast isoprenoids whose synthesis depends on PPS activity and thus clarify why apicoplast biogenesis requires longer-chain PPS activity.

Independent of its specific role in apicoplast biogenesis, PPS function is critical for parasite survival and thus constitutes a new essential arm of isoprenoid metabolism in the apicoplast suitable for development as a therapeutic target. BLAST analysis of the human genome using the PPS protein sequence as query reveals a variety of PPS homologs with modest 20–30% sequence identity to 25–50% of the PPS sequence. The substantial sequence differences with human orthologs will facilitate selective targeting of PPS by chemical inhibitors. Identification of PPS as an apicoplast-targeted enzyme indicates that new metabolic pathways and functions remain to be discovered and/or localized to the apicoplast. These novel functions, which are predicted to be required for organelle maintenance, will enhance our understanding of fundamental apicoplast biology and provide new candidate drug targets for antimalarial therapies.

## Materials and methods

**Key resources table**

| Reagent type (species) or resource | Designation | Source or reference | Identifiers | Additional information |
|---|---|---|---|---|
| Cell line (*Plasmodium falciparum*) | D10 ACPL-GFP | PMID:10775264 | | |
| Cell line (*Plasmodium falciparum*) | NF54-PfMev ACP$_L$-GFP | PMID:32059044 | | |
| Cell line (*Plasmodium falciparum*) | NF54-PfMev ΔDXS | PMID:32815516 | | |
| Cell line (*Plasmodium falciparum*) | Dd2 PPS-RFP (pTyEOE) | This study | | Described in Materials and methods. Can be obtained from Sigala lab. |
| Cell line (*Plasmodium falciparum*) | Dd2 PPS-GFP (pTEOE) | This study | | Described in Materials and methods. Can be obtained from Sigala lab. |
| Cell line (*Plasmodium falciparum*) | Dd2 PPS-HA/FLAG 9xAptamer/ TetR-DOZI | This study | | Described in Materials and methods. Can be obtained from Sigala lab. |

*Continued on next page*

*Continued*

| Reagent type (species) or resource | Designation | Source or reference | Identifiers | Additional information |
|---|---|---|---|---|
| Cell line (*Plasmodium falciparum*) | NF54-PfMev MiaA-KO | This study | | Described in Materials and methods. Can be obtained from Prigge lab. |
| Cell line (*Escherichia coli*) | Top10 pAC-BETAipi | PMID:15659105 | | |
| Software, algorithm | Prism | GraphPad | RRID: SCR_002798 | |
| Chemical compound, drug | Doxycycline | Sigma-Aldrich | Cat. No. D3447 | |
| Chemical compound, drug | Fosmidomycin | Invitrogen Life Technologies | Cat. No. F23103 | |
| Chemical compound, drug | Isopentenyl pyrophosphate | Isoprenoids | Cat. No. IPP001 | |
| Chemical compound, drug | Farnesol | Sigma-Aldrich | Cat. No. F203 | |
| Chemical compound, drug | Geranylgeraniol | Sigma-Aldrich | Cat. No. G3278 | |
| Chemical compound, drug | Decaprenol | Isoprenoids | Cat. No. polyprenol C50 | |
| Chemical compound, drug | DL-mevalonolactone | Cayman Chemicals | Cat. No. 20348 | |
| Chemical compound, drug | Zaragozic acid | Cayman Chemicals | Cat. No.17452 | |
| Chemical compound, drug | MMV019313 | ChemDiv | Cat. No. C498-0579 | |
| Chemical compound, drug | β-Carotene | Sigma-Aldrich | Cat. No. F203 | |
| Antibody | Anti-EF1α (rabbit polyclonal) | PMID:11251817 | | (1:1000) |
| Antibody | Anti-ACP (rabbit polyclonal) | PMID:19768685 | | (1:1000) |
| Antibody | Anti-GFP (3E6) (mouse, monoclonal) | Invitrogen Life Technologies | Cat. No. A11120 | (1:1000) |
| Antibody | Anti-RFP (RF5R) (mouse monoclonal) | Invitrogen Life Technologies | Cat. No. MA5-15257 | (1:1000) |
| Antibody | Anti-GFP (goat, polyclonal) | Abcam | Cat. No. Ab5450 | (1:1000) |
| Antibody | Anti-HA (3F10) (rat, monoclonal) | Roche | Cat. No. 11 867 423 001 | (1:1000) |

## Materials

All reagents were of the highest purity commercially available. The vendor and catalog number are given for individual compounds when first mentioned. Primary antibodies were generally used at 1:1000 dilution and secondary antibodies at 1:10,000 dilution unless specified otherwise.

## Fluorescence microscopy

For live-cell experiments, parasite samples were collected at 38 hr after synchronization with 5% D-sorbitol (Sigma S7900). Parasite nuclei were visualized by incubating samples with 1–2 µg/mL Hoechst 33342 (Thermo Scientific Pierce 62249) for 10–20 min at room temperature. The parasite apicoplast was visualized in D10 (*Waller et al., 2000*) or NF54 mevalonate-bypass (*Swift et al., 2020b*) cells

using the ACP$_L$-GFP expressed by both lines. The parasite mitochondrion was visualized by incubating parasites with 10 nM MitoTracker Red CMXROS (Invitrogen Life Technologies M7512) for 15 min prior to wash-out and imaging. For immunofluorescence assay (IFA) experiments, parasites were fixed, stained, and mounted as previously described (*Tonkin et al., 2004*). For IFA images, the parasite apicoplast was visualized using a polyclonal rabbit anti-ACP antibody (*Gallagher and Prigge, 2010*) and a goat anti-rabbit fluorescent 2° antibody (Invitrogen R37117) and the nucleus was stained with ProLong Gold Antifade Mountant with DAPI (Invitrogen Life Technologies P36931). PPS-RFP was visualized with a mouse monoclonal anti-RFP antibody (Thermo Fisher MA5-15257) and goat anti-mouse fluorescent 2° antibody (Invitrogen A11001). PPS-GFP was detected with a goat anti-GFP antibody (Abcam ab5450) and anti-goat fluorescent 2° antibody. For IFA images of parasites expressing endogenous PPS with C-terminal HA-FLAG tags, apicoplast ACP was visualized with a polyclonal rabbit anti-ACP antibody as above and PPS was visualized using polyclonal rat anti-HA-tag 1° antibody (Roche 3F10) and goat anti-rat (Invitrogen A11006) and donkey anti-goat (Invitrogen A11055) fluorescent 2° antibodies. Images were taken on DIC/bright field, DAPI, GFP, and RFP channels using either a Zeiss Axio Imager or an EVOS M5000 imaging system. Fiji/ImageJ was used to process and analyze images, and intensity plots were generated by the 'plot profile' function using a shared region of interest (identified by white line) on each channel. All image adjustments, including contrast and brightness, were made on a linear scale. For phenotypic analyses, apicoplast morphologies for each experimental condition were assessed for 25 parasites in each of two biological replicate experiments (50 parasites total per condition). Apicoplast morphologies were scored as elongated, focal, or dispersed; counted; and plotted by histogram as the fractional population with the indicated morphology.

## Inhibition and rescue of apicoplast biogenesis

ACP$_L$-GFP D10 and NF54 PfMev parasites were synchronized with 5% (w/v) D-sorbitol for 10 min at room temperature and returned to culture in 10 µM FOS (Invitrogen Life Technologies F23103), 100 nM ATV (Cayman Chemicals 23802), 2 µM DSM1 (*Ganesan et al., 2011*), 6 µM Blast-S (Invitrogen Life Technologies R21001), 5 nM WR99210 (Jacobus Pharmaceuticals), 160 µM ZA/squalestatin (Cayman Chemicals 17452), or 2 µM MMV019313 (ChemDiv C498-0579). For FOS experiments, parasites were left in FOS only or supplemented with 5 µM FOH (Sigma F203), 5 µM GGOH (Sigma G3278), 5 µM decaprenol (Isoprenoids polyprenol C50), 5 µM β-carotene (Sigma C9750), 50 µM DL-mevalonolactone (Cayman Chemicals 20348), or 200 µM IPP (NH$_4^+$ salt, Isoprenoids IPP001). All parasites were cultured for 36 hr after synchronization and then imaged by live-cell fluorescence microscopy to monitor apicoplast status. All concentrations reflect the final concentration in culture medium.

## Parasite synchronization

Parasites were synchronized to the ring stage either by treatment with 5% D-sorbitol (Sigma S7900) or by first magnet-purifying schizonts and then incubating them with uninfected erythrocytes for 5 hr followed by treatment with sorbitol. Results from growth assays and microscopy analyses using either of these synchronization methods were indistinguishable within error, and 5% sorbitol was used unless stated otherwise.

## Delayed mevalonate-rescue assay

NF54 PfMev parasites were synchronized with 5% (w/v) D-sorbitol for 10 min at room temperature and returned to culture in 10 µM FOS. Fifty µM DL-mevalonate was added to cultures immediately or after 30, 34, or 38 hr post-synchronization. Parasitemia was measured by flow cytometry every 24 hr. After 60 hr post-synchronization, parasites from each mevalonate time point were cloned out by limiting dilution. Apicoplast status of all isolated clones was evaluated by live-cell ACP$_L$-GFP fluorescence. ACP$_L$-GFP signal was observed for the presence of distinct branching morphology (apicoplast intact) or the presence of scattered punctate signals throughout the cytosol (apicoplast disruption). A total of 9, 17, 18, and 5 clones from the 0, 30, 34, and 38 hr rescue time points, respectively, were evaluated by microscopy (only five clones returned from the 38 hr rescue time point). Apicoplast (SufB: Pf3D7_API04700) and nuclear (PPS: Pf3D7_0202700) genome PCR (primers 4/5 and 1/2) and mevalonate dependence growth assays were done on two clones from each time point to confirm apicoplast status.

## Parasite culturing and transfection

All experiments were performed using *P. falciparum* Dd2 (*Wellems et al., 1990*), ACP$_L$-GFP D10 (*Waller et al., 2000*), or ACP$_L$-GFP NF54 PfMev (*Swift et al., 2020b*) parasite strains. Parasite strain identities were confirmed on the basis of expected drug resistance and were *Mycoplasma*-free by PCR test. Parasite culturing was performed in Roswell Park Memorial Institute medium (RPMI-1640, Thermo Fisher 23400021) supplemented with 2.5 g/L Albumax I Lipid-Rich BSA (Thermo Fisher 11020039), 15 mg/L hypoxanthine (Sigma H9636), 110 mg/L sodium pyruvate (Sigma P5280), 1.19 g/L HEPES (Sigma H4034), 2.52 g/L sodium bicarbonate (Sigma S5761), 2 g/L glucose (Sigma G7021), and 10 mg/L gentamicin (Invitrogen Life Technologies 15750060). Cultures were generally maintained at 2% hematocrit in human erythrocytes obtained from the University of Utah Hospital blood bank, at 37°C, and at 5% $O_2$, 5% $CO_2$, 90% $N_2$. Parasite-infected erythrocytes were transfected in 1× cytomix containing 50–100 µg midi-prep DNA by electroporation in 0.2 cm cuvettes using a Bio-Rad Gene Pulser Xcell system (0.31 kV, 925 µF). Transgenic parasites were selected on the basis of plasmid resistance cassettes encoding human DHFR (*Fidock and Wellems, 1997*), yeast DHOD (*Ganesan et al., 2011*), or blasticidin-S deaminase (BSD) (*Mamoun et al., 1999*) and cultured in 5 nM WR99210, 2 µM DSM1, or 6 µM Blast-S, respectively. Gene-edited Dd2 parasites that contained PPS (PF3D7_0202700) tagged with the aptamer/TetR-DOZI cassette (*Ganesan et al., 2016*) were maintained in 0.5–1 µM aTc (Cayman Chemicals 10009542). Genetically modified parasites were genotyped by PCR and/or Southern blot, as previously described (*Klemba et al., 2004*). For western blot and IFA studies of PPS-GFP in apicoplast-disrupted Dd2 parasites, transgenic parasites were cultured >7 days in 5 nM WR99210, 1 µM doxycycline (Sigma D9891), and 200 µM IPP to induce stable apicoplast loss prior to parasite harvest.

## Parasite growth assays

Parasite growth was monitored by diluting asynchronous or sorbitol-synchronized parasites to ~0.5% parasitemia and allowing culture expansion over several days with daily media changes. Parasitemia was monitored daily by flow cytometry by diluting 10 µL of each parasite culture well from each of three biological replicate samples into 200 µL of 1.0 µg/mL acridine orange (Invitrogen Life Technologies A3568) in phosphate buffered saline (PBS) and analysis on a BD FACSCelesta system monitoring SSC-A, FSC-A, PE-A, FITC-A, and PerCP-Cy5-5-A channels. Daily parasitemia measurements for asynchronous cultures were plotted as function of time and fit to an exponential growth equation using GraphPad Prism 9.0. For $EC_{50}$ determinations, synchronous ring-stage parasites were diluted to 1% parasitemia and incubated with variable drug concentrations for 48–72 hr without media changes. Parasitemia was determined by flow cytometry in biological triplicate samples for each drug concentration, normalized to the parasitemia in the absence of drug, plotted as a function of the log of the drug concentration (in nM or µM), and fit to a four-parameter dose-response model using GraphPad Prism 9.0.

## Cloning and episomal expression of PPS

The gene encoding PPS (PF3D7_0202700) lacks introns and was cloned by PCR from Dd2 parasite genomic DNA using primers designed for insertion into the XhoI/AvrII sites of pTYEOE (yeast DHOD positive selection cassette) (*Beck et al., 2014*) and pTEOE (human DHFR positive selection cassette) (*Sigala et al., 2015*) vectors in frame with C-terminal RFP and GFP tags, respectively. These vectors are designed to drive episomal protein expression using the HSP86 promoter and for co-transfection with plasmid pHTH that contains the piggyBac transposase (*Balu et al., 2005*) for integration into the parasite genome. A single forward primer was used for PPS cloning into both vectors (primer 1) while reverse primers were vector-specific (primers 2 and 3). All PCR primer sequences are shown in *Supplementary file 1*. Cloning was completed using ligation-independent cloning (QuantaBio RepliQa HiFi Assembly Mix). Cloning products were transformed into Top10 chemically competent cells, and bacterial clones were selected for carbenicillin (Sigma C3416) resistance. Correct plasmid sequence in isolated clonal bacteria was confirmed by both AscI/AatII (NEB) restriction digest and Sanger sequencing (University of Utah DNA Sequencing Core). One-hundred µg of either purified PPS-RFP-TyEOE or PPS-GFP-TEOE in combination with 25 µg of the pHTH transposase plasmid was transfected into Dd2 parasites by electroporation, as described above. Transfected parasites were allowed to expand in the absence of drug for 48 hr before selection with either 2 µM DSM1 or

5 nM WR99210 for PPS-RFP-TyEOE or PPS-GFP-TEOE, respectively. Stable, drug-resistant parasites returned from transfection in 3–6 weeks.

## PPS gene-editing to enable ligand-dependent regulation of protein expression

CRISPR/Cas9-stimulated repair by double-crossover homologous recombination was used to tag the PPS gene (PF3D7_0202700) to encode a C-terminal HA-FLAG epitope tag and the 3' 10× aptamer/TetR-DOZI system (*Ganesan et al., 2016*) to enable regulated PPS expression using aTc. Guide RNA sequences corresponding to TGATATAAAACAAAGTAGCG, CGTGCTAGTTCTATTTTTGC, and GATGATTCAAATAAAAGAAG (primers 6–11) were cloned into a modified version of the previously published pAIO vector (*Spillman et al., 2017*), in which the BtgZI site was replaced with a unique HindIII site to facilitate cloning (primers 12 and 13). To tag the PPS gene, a donor pMG75 (*Ganesan et al., 2016*) repair plasmid was prepared by PCR-amplifying 635 bp of the 3' coding sequence and 679 bp of the 3' untranslated region (UTR) as homology flanks to the PPS gene, fusing these fragments together by PCR with an AflII site in between (679 bp 3' UTR-AflII-635 bp 3' coding sequence), and inserting this fused fragment into the AscI and AatII sites of the pMG75 vector (primers 14–17). A shield mutation was introduced to the 3' end of the coding-sequence homology flank corresponding to the gRNA sequence TGATATAAAACAAAGTAGCG. This mutation (introduced using primer 18) ablated the CRISPR PAM sequence AGG that immediately following the gRNA sequence above by mutating it to AAG, resulting in a silent mutation of the Glu523 codon from GAG to GAA. Sanger sequencing confirmed the correct sequence of the homology flanks inserted into the pMG75 vector. PCR analysis of the final pMG75 vector using primers 39–40 revealed that only nine copies of the aptamer sequence were retained. Before transfection, the pMG74 vector was linearized by AflII digestion performed overnight at 37°C, followed by deactivation with Antarctic Phosphatase (NEB M0289S).

Dd2 parasites were transfected with 50 μg of pAIO Cas9/gRNA vector and 50 μg of the linearized pMG75 donor plasmid, as described above. Parasites were selected on the basis of the BSD resistance cassette encoded by the pMG75 plasmid and returned from transfection after 4–6 weeks. Gene-edited Dd2 parasites resulting from transfection with pAIO Cas9/gRNA-4 (produced with primers 10/11) contained PPS (PF3D7_0202700) tagged with the aptamer/TetR-DOZI cassette (*Ganesan et al., 2016*) and were maintained in 0.5–1 μM aTc (Cayman Chemicals 10009542). Genetically modified parasites were genotyped by Southern blot, as previously described (*Klemba et al., 2004*). Briefly, genomic DNA from the polyclonal parasites that returned from transfection was digested with BamHI and SpeI (New England Biolabs) and transferred to membrane (Nytran SuPerCharge) using the TurboBlotter system (VWR 89026–838). A DNA probe consisting of the 5 750 bp of the PPS gene was produced by PCR (primers 16/17). Probe labeling, hybridization, and visualization were performed using the AlkPhos Direct Labeling and Detection System (VWR 95038-288) and CDP-Star reagent (VWR 95038-292). The Southern blot confirmed complete integration into the PPS locus without evidence for unmodified parasites, and the polyclonal parasites were used for all subsequent experiments.

## Analysis of PPS transcript levels

Biological replicate cultures of PPS aptamer/TetR-DOZI parasites were synchronized in 5% D-sorbitol and grown for 72 hr in +aTc (four biological replicates), -aTc (four biological replicates), or -aTc/+ IPP (200 μM) (two biological replicates) conditions prior to harvest. Four mL cultures at ~10% were harvested by centrifugation (2000 rpm for 3 min) and stored at –20°C until use. Total RNA was isolated from frozen parasite-infected blood pellets using a modified Trizol (Invitrogen) extraction protocol. Five mL Trizol (Invitrogen) was added to thawed pellets on ice, pipetted 20–30 times to resuspend, and pulse-vortexed 20 times for 15 s. Two mL chloroform was added to each sample and vortexed, incubated on ice for 5 min, then spun for 10 min at 4°C at 5000 rpm without brake. The top, aqueous layer (~3 mL) was transferred to a new tube. Five mL of isopropanol was added to each sample, gently mixed, and incubated at –80°C for 20 min or –20°C overnight. Samples were spun at 5000 rpm for 30 min, washed with freshly made solution of 70% ethanol, then spun again for 10 min. Ethanol was removed and pellets were dried 30 min on ice. RNA pellets were resuspended in RNAse-free water, quantitated, and used immediately or stored at –80°C. One μg of RNA was DNAse-treated and reverse-transcribed using Superscript IV kit (Invitrogen) with the addition of gene-specific reverse primers 31–38. Subsequent cDNA was analyzed in technical duplicate through qPCRs with

SYBR Green fluorescent probe (Invitrogen) in a Roche Lightcycler. Cp values for PPS (primers 35–36) were normalized to the average of two nuclear-encoded control genes (I5P, PF3D7_0802500; ADSL, PF3D7_0206700; primers 31–34), then used to calculate relative RNA abundance values for each culture condition, with the +aTc sample normalized to a value of 1. Data is reported as average ± SD of replicates for each condition.

## Synchronous growth assays of PPS KD parasites

Dd2 parasites tagged at the genomic PPS locus with the aptamer/TetR-DOZI system were synchronized by 5% D-sorbitol to ring-stage parasites and allowed to expand ±aTc in two or three biological replicate samples. Parasitemia values were measured daily by flow cytometry and plotted as the average ± SD of replicate samples. For growth-rescue experiments, synchronous parasites were allowed to expand ±aTc, and -aTc plus 200 µM IPP, 5 µM FOH, 5 µM GGOH, or 5 µM decaprenol ($C_{50}$-OH). For growth-rescue experiments involving FOS, PPS KD parasites were synchronized to rings with 5% D-sorbitol and grown for 4 days (96 hr) ± aTc. After 96 hr, all culture wells were synchronized again with 5% D-sorbitol and supplemented with 10 µM FOS and 200 µM IPP, 5 µM FOH, 5 µM GGOH, or 5 µM decaprenol. Parasites were cultured for another 38 hr before harvest at 134 total hours post-initial synchronization for IFA analysis of apicoplast morphology. Parasites grown ±aTc with 10 µM FOS and 200 µM IPP were allowed to expand for an additional 48 hr and harvested at 182 hr post-initial synchronization for analysis by IFA and qPCR for apicoplast morphology and apicoplast:nuclear genome levels, respectively.

## qPCR analysis of apicoplast:nuclear genomic DNA levels

Genomic DNA was extracted from triplicate parasite samples grown ±aTc with 10 µM FOS and 200 µM IPP and harvested at 182 hr post-initial synchronization. DNA extraction was performed using the QIAmp DNA Blood Mini Kit (Qiagen 51104). Primers for qPCR were designed to amplify a 120–140 bp region of an apicoplast gene (TufA, PF3D7_API02900, primers 35–36) and each of two nuclear genes (I5P, PF3D7_0802500; ADSL, PF3D7_0206700; primers 31–34). Approximately 100 ng of DNA was amplified in each of three biological replicates with PowerUp SYBR Green Master Mix (Thermo Fisher A25741) in a 96-well plate with 20 µL reaction volume on a Quantstudio3 Real Time PCR system. Specificity of primer amplification was confirmed for every sample by identifying only one melting temperature for the product of each qPCR reaction. Abundance of apicoplast relative to nuclear DNA was determined by comparative Ct analysis (*Schmittgen and Livak, 2008*), with amplification of TufA (apicoplast) and I5P (nuclear) and calculation of $2^{\Delta Ct}$, where $\Delta Ct = Ct_{TufA} - Ct_{I5P}$. As a positive control, abundance of a second nuclear gene (ADSL) relative to I5P was calculated similarly. The $2^{\Delta Ct}$ value for TufA or ADSL was normalized to +aTc for each gene to determine a normalized target gene:control gene DNA abundance. Error bars represent the standard deviation between replicates, and p values were determined by two-tailed unpaired t-test in GraphPad Prism 9.0.

## MiaA gene disruption

The gene encoding MiaA (PF3D7_1207600) was disrupted in the NF54 PfMev line using CRISPR/Cas9 and gene deletion by double-crossover homologous recombination, similar to the recently described disruption of the DXPR gene (PF3D7_1467300) (*Swift et al., 2020b*). Homology arm regions (411 bp for the 5' arm and 540 bp for the 3' arm) were PCR-amplified from genomic DNA with primers 19–22 and cloned into the vector pRS (*Swift et al., 2020b*) using ligation-independent cloning (In-Fusion, Clontech, Mountain View, CA). A guide RNA with sequence AATAACGATATTAAATGTAA was cloned into a modified pAIO vector called pCasG (*Rajaram et al., 2020*) using primers 23 and 24; 75 µg of pRS-miaA-KO plasmid was combined with 75 µg of the pCasG guide RNA plasmid and transfected into NF54 PfMev parasites. Transfected parasites were allowed to expand for 48 hr in 50 µM mevalonate before selection with 5 nM WR99210 and 50 µM mevalonate. Parasites returning from positive selection were genotyped by PCR using primers 25–30. Asynchronous growth of ΔMiaA PfMev parasites ± Mev compared to parental PfMev parasites was performed on biological duplicate samples. Average parasitemia values ± SD were plotted versus time and fit to an exponential growth equation in GraphPad Prism 9.0. Apicoplast (SufB: Pf3D7_API04700) and nuclear (LDH: PF3D7_1324900) genome PCR was performed to confirm apicoplast status in parental PfMev and ΔMiaA parasites, as previously reported (*Swift et al., 2020b*).

## Western blots

Samples of episomal PPS-GFP Dd2 or endogenously HA-FLAG-tagged PPS Dd2 parasites were harvested by centrifugation and treated with 0.05% saponin (Sigma 84510) in PBS for 5 min at room temperature and spun down by centrifuge at 5000 rpm for 30 min at 4°C; 2% SDS or LDS sample buffer (Life Technologies NP0007) was added to saponin pellets, resuspended by sonication, and incubated overnight at 4°C. 5× Sample buffer containing beta-mercaptoethanol was added to parasite samples before heating at 95°C for 5 min and centrifuging at 13,000 rpm for 5 min. Samples were fractionated by SDS-polyacrylamide gel electrophoresis using 10% acrylamide gels run at 120 V in the BIO-RAD mini-PROTEAN electrophoresis system. Fractionated proteins were transferred from polyacrylamide gel to a nitrocellulose membrane at 100 V for 1 hr using the BIO-RAD wet transfer system. Membranes were blocked in 1% casein/PBS for 1 hr at room temperature and then probed with primary antibody overnight at 4°C and secondary antibody at room temperature for 1 hr. Episomal PPS-RFP parasite samples were probed with 1:1000 mouse anti-RFP (Invitrogen Life Technologies MA5-15257) and 1:10,000 donkey anti-mouse DyLight800 (Invitrogen Life Technologies SA5-10172). Endogenous HA-FLAG-tagged PPS parasite samples were probed with Roche rat anti-HA monoclonal 3F10 (Sigma 11867423001) and mouse anti-FLAG (Sigma F1804) primary antibodies and goat anti-rat IRDye800CW (Licor 926–32219) and goat anti-mouse IRDye800CW (Licor 925–32210) secondary antibodies. As a loading control, blots were probed with rabbit anti-EF1α 1° antibody (*Mamoun and Goldberg, 2001*) and donkey anti-rabbit-IRDye680 2° antibody (Licor 926–68023). Densitometry analysis of band intensities was performed using the Licor Image Studio Lite software and median local background.

## Sequence similarity analysis and structural homology modeling

Sequence similarity searches for *P. falciparum* homologs to chicken FPPS (Uniprot P08836) were performed by BLASTP analysis as implemented at the Plasmodium Genomics Resource webpage (https://www.plasmodb.org/). Sequence similarity searches using the PPS (PF3D7_0202700) protein sequence as query were carried out using NCBI BLAST (*Boratyn et al., 2013*) (excluding organisms in the phylum Apicomplexa to which *P. falciparum* belongs) and MPI HHpred (*Zimmermann et al., 2018*). A homology model of PPS was generated by the MPI HHpred software using the X-ray crystallographic structural model of *E. coli* OPPS (PDB 3WJK), which was one of the top 10 homology hits by HHpred analysis, as template. Structural models were visualized using PyMol (Schrödinger).

## β-Carotene extraction and analysis by mass spectrometry

For determination of β-carotene levels in parasite-infected versus uninfected erythrocytes, 35 mL of 4% hematocrit *P. falciparum* culture infected at 13–15% parasitemia with the Dd2 PPS aptamer/TetR-DOZI KD parasites were collected after 6 days of growth in the presence or absence of 1 μM aTc. Uninfected erythrocyte samples were prepared by collecting 20 mL of 4% hematocrit uninfected culture incubated for 6 days in RPMI media that lacked or contained 2.5 g/L AlbuMAX. Samples of infected or uninfected erythrocytes were harvested by centrifugation, lysed by 0.05% saponin, and pelleted by centrifugation. Saponin pellets were washed in PBS and then extracted three times in 1 mL of chilled acetone (pellet was briefly sonicated after addition of the first acetone volume). The supernatant of each extraction was pooled and dried down by vacuum concentration (Speed Vac). Three biological replicates of each sample were prepared. For analysis of AlbuMAX, 85 mg of dry AlbuMAX (equivalent to the AlbuMAX content in 35 mL of complete culture media) was extracted in 3 volumes of cold acetone, and supernatants were combined and dried as above.

For analysis of β-carotene synthesis, NF54 PfMev parasites were cultured and expanded over three intraerythrocytic cycles in media containing 50 μM 2-$^{13}$C-mevalonate and 10 μM FOS. This strategy was chosen to inhibit MEP pathway activity, ensure full $^{13}$C-labeling of the endogenous IPP and DMAPP precursor pool within parasites produced by the cytoplasmic bypass enzymes, and result in a distinguishable 8 Da mass increase for any β-carotene derived from de novo synthesis. Final parasite samples contained 70 mL of culture at 15% parasitemia and were collected by centrifugation prior to 0.05% saponin lysis, centrifugation, and washing the pellet in PBS. The final pellet was extracted in acetone and dried, as described above.

As a positive control for detecting isotopic incorporation of $^{13}$C-labeled precursors into biosynthesized β-carotene, we turned to studies of *E. coli* bacteria engineered to biosynthesize β-carotene

(**Cunningham and Gantt, 2005**) and grown in minimal M9 medium with uniformly labeled $^{13}$C-glucose as the sole carbon source. Growth of bacteria in these conditions was expected to lead to a 40 Da mass increase in detected β-carotene. Five mL cultures of pAC-BETAipi *E. coli* or untransformed Top10 *E. coli* were allowed to expand over 2 days at 30°C in the dark. Bacterial cultures were harvested by centrifugation at 5000 rpm for 10 min, and bacterial pellets were extracted three times in cold acetone and dried, as described above.

LC-MS-grade methanol, acetonitrile, isopropyl alcohol, chloroform, and formic acid were purchased from VWR. Samples were resuspended in 50 μL MeOH/CHCl$_3$ (2 mM LiI), and a sample volume of 10 μL was injected onto a Phenomenex Luna 150 × 2.1 mm reverse-phase C8 column maintained at 30°C and connected to an Agilent HiP 1290 Sampler, Agilent 1290 Infinity pump, and Agilent 6545 Accurate Mass Q-TOF dual AJS-ESI mass spectrometer. The instrument was operated in positive ion mode, and the source gas temperature was 275°C with a drying gas flow of 12 L/min, nebulizer pressure of 35 psig, sheath gas temperature of 325°C, and sheath gas flow of 12 L/min. VCap voltage was set at 3500 V, nozzle voltage 250 V, fragmentor at 90 V, skimmer at 65 V, octopole RF peak at 750 V, and a scan range *m/z* 40–900. The mobile solvent phase A was H$_2$O with 0.1% formic acid, and mobile phase B was MeOH:ACN:IPA (2:2:1 v/v) with 0.1% formic acid. The chromatography gradient started at 80% mobile phase B then increased to 100% B over 6 min where it was held until 9.9 min and then returned to the initial conditions and equilibrated for 5 min. The column flow rate was 0.5 mL/min.

Results from LC-MS experiments were collected using Agilent Mass Hunter (MH) Workstation and analyzed using the software packages MH Qual and MH Quant (Agilent Technologies, Inc). Unlabeled, 2-$^{13}$C-mevalonate-labeled, and uniform $^{13}$C-glucose-labeled β-carotene were analyzed using the molecular ions of *m/z* 536.4377, *m/z* 544.4645, and *m/z* 576.5718, respectively. Fragmentation profiling of unlabeled β-carotene by MS/MS confirmed the expected product ions at *m/z* 444 and *m/z* 119, as previously reported (**Rivera et al., 2014**). For quantitation of unlabeled β-carotene levels in experimental samples and determination of a limit of detection (LOD), a calibration curve was constructed using serial dilutions of commercial β-carotene (Sigma C9750). The concentration LOD for a 10 μL sample of unlabeled β-carotene in this assay was 2.6 ng/mL. Integrated peak areas for unlabeled β-carotene in experimental samples were converted to concentration values in the 10 μL sample using this calibration curve.

## Acknowledgements

We thank Belén Cassera, James Cox, Dale Poulter, and members of the Sigala lab for helpful discussions. PAS holds a Career Award at the Scientific Interface from the Burroughs Wellcome Fund and is a Pew Scholar in the Biomedical Sciences, supported by The Pew Charitable Trusts. STP is supported by the Johns Hopkins Malaria Research Institute and the Bloomberg Philanthropies. DNA synthesis and sequencing, epifluorescence microscopy, MS metabolomics, generation of CRISPR/Cas9 reagents, and flow cytometry were performed using core facilities at the University of Utah.

## Additional information

### Funding

| Funder | Grant reference number | Author |
| --- | --- | --- |
| National Institute of General Medical Sciences | R35GM133764 | Paul A Sigala |
| Congressionally Directed Medical Research Programs | W81XWH1810060 | Paul A Sigala |
| National Institute of Allergy and Infectious Diseases | R01AI125534 | Sean T Prigge |
| Burroughs Wellcome Fund | 1011969 | Paul A Sigala |
| Pew Charitable Trusts | 32099 | Paul A Sigala |

| Funder | Grant reference number | Author |
|---|---|---|
| National Institute of Diabetes and Digestive and Kidney Diseases | T32DK007115 | Megan Okada Amanda Mixon |
| National Institute of Allergy and Infectious Diseases | T32AI007417 | Krithika Rajaram |
| National Institute of Diabetes and Digestive and Kidney Diseases | U54DK110858 | John Alan Maschek |
| National Institutes of Health | 1S10OD016232 | John Alan Maschek |
| National Institutes of Health | 1S10OD018210 | John Alan Maschek |
| National Institutes of Health | 1S10OD021505 | John Alan Maschek |
| Bloomberg Philanthropies | | Sean T Prigge |
| Johns Hopkins Malaria Research Institute | | Sean T Prigge |

The funders had no role in study design, data collection and interpretation, or the decision to submit the work for publication.

### Author contributions

Megan Okada, Formal analysis, Investigation, Methodology, Validation, Visualization, Writing - original draft, Writing - review and editing; Krithika Rajaram, Formal analysis, Investigation, Methodology, Validation, Writing - review and editing; Russell P Swift, Investigation, Methodology; Amanda Mixon, John Alan Maschek, Investigation, Methodology, Writing - review and editing; Sean T Prigge, Conceptualization, Funding acquisition, Methodology, Project administration, Supervision, Writing - review and editing; Paul A Sigala, Conceptualization, Funding acquisition, Methodology, Project administration, Supervision, Validation, Writing - original draft, Writing - review and editing

### Author ORCIDs

Megan Okada http://orcid.org/0000-0003-4398-9819
Krithika Rajaram http://orcid.org/0000-0003-4830-5471
Sean T Prigge http://orcid.org/0000-0001-9684-1733
Paul A Sigala http://orcid.org/0000-0002-3464-3042

### Decision letter and Author response

Decision letter https://doi.org/10.7554/eLife.73208.sa1
Author response https://doi.org/10.7554/eLife.73208.sa2

## Additional files

### Supplementary files
• Transparent reporting form
• Supplementary file 1. Primers used in this study.

### Data availability
All data generated or analyzed during this study are included in the manuscript and supporting files. Figure 1- source data 1 contains the numerical scoring data for all microscopy analyses.

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
