## [Editor Report]

This is an excellent, innovative and high quality study that reveals an essential role for isoprenoids within the *Plasmodium* apicoplast, and demonstrates a likely polyprenol synthase that is required for apicoplast biogenesis. This is an important finding for understanding apicoplast and isoprenoid biology in general, and is significant because synthesis of isoprenoids appears to be the only essential role for the apicoplast in asexual intraerythrocytic stages.

---

## [Decision Letter]

**Decision letter after peer review:**

Thank you for submitting your article "Critical Role for Isoprenoids in Apicoplast Biogenesis by Malaria Parasites" for consideration by *eLife*. Your article has been reviewed by 3 peer reviewers, and the evaluation has been overseen by a Reviewing Editor and Dominique Soldati-Favre as the Senior Editor. The following individuals involved in review of your submission have agreed to reveal their identity: Stuart A Ralph (Reviewer #1); Taco W.A. Kooij (Reviewer #3).

Overall the reviewers are aligned in considering the work suitable for publication at *eLife* however there are some issues that need to be addressed.

Essential revisions:

1. Perform an IPP rescue on knockdown of the PPS

2. Provide experimental evidence that PPS is inside the apicoplast and not associated outside of the organelle. As it stands the model presented in figure 9 is not supported by the data.The reviewers are asking for more data (bioinformatics or wet lab) to explain this unusual phenomenon and to adjust the figure 9 and conclusion accordingly.

3. Determine if decaprenol indeed rescues growth

4. Fix the issues of statistics by including triplicates. For some of the data it would be sufficient to remove statistical assertions, but for the growth assays triplicate is the minimal standard.

5. Fig6B and C include western blot showing lower PPS levels upon aTc wash-out.

*Reviewer #1 (Recommendations for the authors):*

The authors use some larger prenols than have been previously used to interrogate isoprenoid utilisation – they report that decaprenol (50 carbons) provided observed rescue of apicoplast branching. There isn't any mention of whether this also rescued growth. Were those experiments done? I would not suggest any new work if they haven't, but would be nice to know if these have similar impact to either IPP or GGOH on growth (i.e long term or temporary rescue, respectively)

Lone 479 – Did the authors use the Flag (HA) tagged chromosomal copy of the PPS to perform subcellular localisation of the enzyme?

Line 484 – looks like only the mature version is apparent here, not the version including a transit peptide. This isn't surprising if the GFP-fusion version in figure is a bit overexpressed in comparison, but might need an extra sentence of explanation to explain the difference between the westerns in Figure 5 and 6.

Figure 9 – Heme a should use a capital A.

*Reviewer #2 (Recommendations for the authors):*

In several cases, statistical analyses and presented data reflect two independent experiments (Figure 7C, 4B, 1C). Three should be the norm.

Line 208-210: The authors refer to Figure 1 and discuss rescue through IPP. However, the Figure and the supplementary data referred to only show data of PfMev parasites rescued through Mev rather than IPP.

*Reviewer #3 (Recommendations for the authors):*

While the conclusions remain unaltered, in my view the use of statistics on all the examples where data on individual parasites from only two independent experiments were collected is incorrect. In order to be able to perform statistics correctly a third biological replicate should be analysed for all these experiments and the means for these should then be compared instead. However, I don't think you need these statistics as the results are very clear-cut and it would be a lot of work to repeat all this work. This applies for many figures, I might have missed some still, but at least: Figure 1C,E,S2, Figure 4B,S1,S3, Fig5S2, Fig7C,E,F: even 1? experiment for 7E.

Figure 2B: It is unclear whether the parasitemias are the average of all or selected clones.

Figure 2C: The way the data are presented now, it appears to suggest that following late treatment there are only or at least a vast majority of clones with disrupted apicoplast. This is clearly not the case as shown in 2D. Therefore, I think it would be more correct to show an example of intact and disrupted for all conditions in the main figure, or at least show the major type of clone observed – which would be intact for the 30h and even 34h treatment samples. If choosing to show examples of intact and disrupted, the authors could consider to integrate the data from 2D by including the numbers and percentages next to the images. Furthermore, the growth data are from a single experiment, so even though "error bars are smaller than the data points", error bars are irrelevant here. As the experiment is merely a confirmation of the expected phenotype there is no need for additional experiments in my humble opinion.

Figure 5: Personally I prefer single colour panels to be shown in grayscale for enhanced dynamic range of the human eye, though admittedly the images are very clear and converting all is not worth the effort. However, the colour blind it would be nice to use green-magenta instead of green-red. This will immediately deal with the possible confusion of showing both GFP and RFP in green.

Fig6B and C: I'm not sure why "inconsistencies" prevented detection of PPS with or without aTc, when it was detected in B. It would be nice to include a Western blot showing lower PPS levels upon aTc wash-out.

---

## [Author Response]

Essential revisions:1. Perform an IPP rescue on knockdown of the PPS

We currently show in Figure 6E that culture supplementation with 200 µM IPP rescues parasite growth from downregulation of PPS expression in -aTc conditions. To address R1’s concern that the PPS transcript reduction in -aTc conditions in Figure 6D could be due to non-specific defects in parasite fitness from loss of PPS function, we have repeated the RT-qPCR experiment in the presence of 200 µM IPP to rescue parasites and decouple their viability from PPS knockdown. We observe identical reduction of PPS transcript levels in -aTc conditions (relative to control gene) in the presence or absence of exogenous IPP, supporting our conclusion that PPS transcripts are selectively degraded in -aTc conditions. We also performed western blot experiments of PPS knockdown in -aTc +IPP conditions to avoid non-specific effects from impaired parasite fitness in -aTc conditions alone.

2. Provide experimental evidence that PPS is inside the apicoplast and not associated outside of the organelle. As it stands the model presented in figure 9 is not supported by the data.The reviewers are asking for more data (bioinformatics or wet lab) to explain this unusual phenomenon and to adjust the figure 9 and conclusion accordingly.

We agree that apicoplast targeting by PPS has somewhat unusual features in that the N-terminus is not recognized as a signal peptide by SignalP (versions 3-5), despite clear recognition by PlasmoAP of a transit peptide sequence. Nevertheless, multiple experimental observations strongly support our conclusion that PPS is targeted to and imported into the apicoplast. These observations include co-localization of PPS with apicoplast-targeted ACP, dispersal of PPS signal after apicoplast disruption upon culture in IPP/doxycycline, and most importantly western blot detection of both a precursor and N-terminally processed mature form of PPS in untreated parasites but exclusive detection of the precursor protein upon apicoplast disruption. We have also added new localization data to Figure 6 for the endogenous HA-tagged PPS, showing co-localization with apicoplast ACP (including detailed signal overlap analysis via 2-dimensional intensity versus distance plot).

Because our detection of PPS expression relies on a C-terminal epitope, detection of a truncated mature form implies processing at the N-terminus. To our knowledge, N-terminal processing of an apicoplast-targeted protein is an exclusive property of proteins imported into the organelle. Although multiple proteins have been shown or suggested to associate with the apicoplast exterior or target the outer apicoplast membrane (e.g., ATG8, oTPT, and FTSH1), none of these proteins is N-terminally processed (see Pubmed IDs: 22900071, 24025672, 28826494, and 16760253). In all cases, these proteins are either unprocessed (ATG8 and oTPT) or subject to C-terminal processing (FTSH1). N-terminal processing of PPS is therefore strongly suggestive of import into the apicoplast. Loss of such N-terminal processing in apicoplast-disrupted parasites is consistent with a model that such processing requires apicoplast import.

We agree that N-terminal processing does not specify the sub-compartment inside the apicoplast to which PPS is localized, which could include the matrix and/or multiple membranes or inter-membrane spaces. Indeed, N-terminal processing has been reported for both matrix and membrane-targeted proteins imported into the apicoplast (e.g., Pubmed 16760253). The central conclusions of our manuscript are not affected by this ambiguity. For simplicity, we have depicted PPS localization in Figure 9 as within the apicoplast matrix but agree that more work is required to specify sub-compartmental targeting within the apicoplast. We have added a paragraph to the Discussion to explore PPS localization and to explicitly state that further experiments will be needed to specify targeting of PPS within the apicoplast. We have also modified the legend for Figure 9 to explicitly state that depiction of PPS localization in the matrix is a model and that additional experiments are required to test and refine this localization.

As a point of perspective, we note that multiple proteins detected in the Boucher et al. apicoplast proteome (Pubmed 30212465) lack an identifiable signal peptide by SignalP yet are clearly imported into the apicoplast. These proteins include the key MEP pathway enzymes DXR (PF3D7_1467300) and IspD (PF3D7_0106900), holo ACP synthase (PF3D7_0420200), FabB/F (PF3D7_0626300), and the E1 subunit of pyruvate dehydrogenase (PF3D7_1446400). Thus, apicoplast import despite lack of identifiable signal peptide by SignalP is not unique to PPS but general to multiple (if not many) apicoplast-targeted proteins. These observations suggest to us that protein N-termini in *Plasmodium* can have sequence properties compatible with apicoplast targeting that are broader and more heterogenous than ER-targeting sequences from other eukaryotic organisms that comprise the training sets upon which SignalP is currently based. It remains a future challenge to fully understand these N-terminal properties and their relation to apicoplast targeting and import in *Plasmodium*. We have added a section to the Discussion to explore these considerations.

3. Determine if decaprenol indeed rescue growth

We currently show in Figure 6E that culture supplementation with 10 µM decaprenol rescues parasite growth in -aTc conditions and that the magnitude of rescue by decaprenol is indistinguishable from IPP. In Figure 7B and 7C, we show that decaprenol is the only isoprenoid that rescues apicoplast biogenesis from PPS knockdown in -aTc (+FOS) conditions.

4. Fix the issues of statistics by including triplicates. For some of the data it would be sufficient to remove statistical assertions, but for the growth assays triplicate is the minimal standard.

We have repeated all growth assays and now include data derived from biological triplicates. We have removed p-values for microscopy experiments.

5. Fig6B and C include western blot showing lower PPS levels upon aTc wash-out.

We have added western blot data in the revised Figure 6 showing decreased PPS expression in -aTc +IPP conditions relative to a loading control.

Reviewer #1 (Recommendations for the authors):The authors use some larger prenols than have been previously used to interrogate isoprenoid utilisation – they report that decaprenol (50 carbons) provided observed rescue of apicoplast branching. There isn't any mention of whether this also rescued growth. Were those experiments done? I would not suggest any new work if they haven't, but would be nice to know if these have similar impact to either IPP or GGOH on growth (i.e long term or temporary rescue, respectively)

Figure 6E shows that exogenous decaprenol rescues parasite growth due to PPS knockdown in -aTc conditions for at least 5 days (relative to -aTc alone), and that the magnitude of rescue by decaprenol is similar to IPP.

Line 479 – Did the authors use the Flag (HA) tagged chromosomal copy of the PPS to perform subcellular localisation of the enzyme?

We have added new images and analyses in Figure 6 showing co-localization of the endogenous, HA-FLAG-tagged PPS with apicoplast ACP.

Line 484 – looks like only the mature version is apparent here, not the version including a transit peptide. This isn't surprising if the GFP-fusion version in figure is a bit overexpressed in comparison, but might need an extra sentence of explanation to explain the difference between the westerns in Figure 5 and 6.

We have followed the reviewer’s suggestion and added a sentence to clarify this difference.

Figure 9 – Heme a should use a capital A.

Corrected.

Reviewer #2 (Recommendations for the authors):In several cases, statistical analyses and presented data reflect two independent experiments (Figure 7C, 4B, 1C). Three should be the norm.

We have removed the statistical analyses from these figure panels.

Line 208-210: The authors refer to Figure 1 and discuss rescue through IPP. However, the Figure and the supplementary data referred to only show data of PfMev parasites rescued through Mev rather than IPP.

We thank the reviewer for pointing out this error. Figure 1—figure supplements 2 and 3 show the results of IPP rescue of FOS treatment of D10 parasites. We have corrected the Figure reference in this section.

Reviewer #3 (Recommendations for the authors):While the conclusions remain unaltered, in my view the use of statistics on all the examples where data on individual parasites from only two independent experiments were collected is incorrect. In order to be able to perform statistics correctly a third biological replicate should be analysed for all these experiments and the means for these should then be compared instead. However, I don't think you need these statistics as the results are very clear-cut and it would be a lot of work to repeat all this work. This applies for many figures, I might have missed some still, but at least: Figure 1C,E,S2, Figure 4B,S1,S3, Fig5S2, Fig7C,E,F: even 1? experiment for 7E.

We have followed the reviewer’s suggestions. For growth assays, we have performed a third biological replicate and updated those figures and the indicated statistical analyses. For microscopy experiments, we have removed the P values.

Figure 2B: It is unclear whether the parasitemias are the average of all or selected clones.

The growth assays shown correspond to the specific clone labeled to the left of each graph. We have modified the figure legend to clarify that the growth assay corresponds to that single clone but is representative of the multiple clones that were studied. Microscopy images of apicoplast morphology for all clones are included in Figure 2—figure supplements 1-4.

Figure 2C: The way the data are presented now, it appears to suggest that following late treatment there are only or at least a vast majority of clones with disrupted apicoplast. This is clearly not the case as shown in 2D. Therefore, I think it would be more correct to show an example of intact and disrupted for all conditions in the main figure, or at least show the major type of clone observed – which would be intact for the 30h and even 34h treatment samples. If choosing to show examples of intact and disrupted, the authors could consider to integrate the data from 2D by including the numbers and percentages next to the images. Furthermore, the growth data are from a single experiment, so even though "error bars are smaller than the data points", error bars are irrelevant here. As the experiment is merely a confirmation of the expected phenotype there is no need for additional experiments in my humble opinion.

We thank the reviewer for this suggestion and agree on the importance of clearly displaying all data. For clarity and because the presence of clones with a disrupted apicoplast is the more revealing and thus important result, we have only included data for each time point in Figure 2C for a parasite clone with disrupted apicoplast if such disruption was observed. We note that microscopy images of all clones are shown in Figure 2—figure supplements 1-4 and are summarized in Figure 2D. Rather than cluttering Figure 2C with additional images and percentages, we have modified Figure 2D to include the percentage of clonal parasites with intact or disrupted apicoplast. We have also modified the Figure legend to explicitly refer the reader to figure supplements for images of all clones.

Figure 5: Personally I prefer single colour panels to be shown in grayscale for enhanced dynamic range of the human eye, though admittedly the images are very clear and converting all is not worth the effort. However, the colour blind it would be nice to use green-magenta instead of green-red. This will immediately deal with the possible confusion of showing both GFP and RFP in green.

We have modified Figure 5 to consistently depict GFP fluorescence in green and RFP in red, to avoid confusion. We also note that the merge panels depict the strong single-color overlap between PPS-RFP and apicoplast ACP signals as well as the clearly non-overlapping signals for MitoTracker and PPS-GFP.

Fig6B and C: I'm not sure why "inconsistencies" prevented detection of PPS with or without aTc, when it was detected in B. It would be nice to include a Western blot showing lower PPS levels upon aTc wash-out.

We now provide a western blot that shows loss of PPS expression for parasites grown in -aTc +IPP conditions, with appropriate loading control.